# Modeling and simulation of neocortical micro- and mesocircuitry (Part I, anatomy)

**Michael W Reimann**[1*†‡], **Sirio Bolaños-Puchet**[1*†‡], **Jean-Denis Courcol**[1†‡], **Daniela Egas Santander**[1†‡], **Alexis Arnaudon**[1], **Benoît Coste**[1], **Fabien Delalondre**[1], **Thomas Delemontex**[1], **Adrien Devresse**[1], **Hugo Dictus**[1], **Alexander Dietz**[1], **András Ecker**[1], **Cyrille Favreau**[1], **Gianluca Ficarelli**[1], **Mike Gevaert**[1], **Joni Herttuainen**[1], **James B Isbister**[1], **Lida Kanari**[1], **Daniel Keller**[1], **James King**[1], **Pramod Kumbhar**[1], **Samuel Lapere**[1], **Jānis Lazovskis**[2], **Huanxiang Lu**[1], **Nicolas Ninin**[1], **Fernando Pereira**[1], **Judit Planas**[1], **Christoph Pokorny**[1], **Juan Luis Riquelme**[1], **Armando Romani**[1], **Ying Shi**[1], **Jason P Smith**[3], **Vishal Sood**[1], **Mohit Srivastava**[4], **Werner Van Geit**[1], **Liesbeth Vanherpe**[1], **Matthias Wolf**[1], **Ran Levi**[5§], **Kathryn Hess**[6§], **Felix Schürmann**[1§], **Eilif B Muller**[1§], **Henry Markram**[1*§], **Srikanth Ramaswamy**[1,7*§]

[1]Blue Brain Project, École polytechnique fédérale de Lausanne (EPFL), Campus Biotech, Geneva, Switzerland; [2]Riga Business School, Riga Technical University, Riga, Latvia; [3]Nottingham Trent University, Nottingham, United Kingdom; [4]ELKH-University of Debrecen, Neuroscience Research Group, Debrecen, Hungary; [5]University of Aberdeen, Aberdeen, United Kingdom; [6]Laboratory for Topology and Neuroscience (UPHESS), Brain Mind Institute, School of Life Sciences, École polytechnique fédérale de Lausanne (EPFL), Lausanne, Switzerland; [7]Neural Circuits Laboratory, Newcastle University, Newcastle, United Kingdom

**\*For correspondence:**
mwr@reimann.science (MWR);
sirio.bolanospuchet@epfl.ch (SB-P);
henry.markram@epfl.ch (HM);
Srikanth.Ramaswamy@newcastle.ac.uk (SR)

†These authors contributed equally to this work

‡Co-lead authors

§Co-senior authors

## eLife Assessment

This manuscript reports a detailed model of juvenile rat somatosensory cortex, consisting of 4.2 million morphologically and biophysically detailed neuron models, arranged in space and connected according to diverse experimental data - a **valuable** tool for the field. The construction of the model is based on a methodology with **solid** supporting evidence. It should be noted that, by necessity, such a large-scale model development involves many assumptions, interpolations, and decisions that could have compounding downstream effects on further analyses that may be difficult to disambiguate.

**Abstract** The function of the neocortex is fundamentally determined by its repeating microcircuit motif, but also by its rich, interregional connectivity. We present a data-driven computational model of the anatomy of non-barrel primary somatosensory cortex of juvenile rat, integrating whole-brain scale data while providing cellular and subcellular specificity. The model consists of 4.2 million morphologically detailed neurons, placed in a digital brain atlas. They are connected by 14.2 billion synapses, comprising local, mid-range and extrinsic connectivity. We delineated the limits of determining connectivity from neuron morphology and placement, finding that it reproduces targeting by Sst+ neurons, but requires additional specificity to reproduce targeting by PV+ and VIP+ interneurons. Globally, connectivity was characterized by local clusters tied together through hub neurons in layer 5, demonstrating how local and interregional connectivity are complicit, inseparable networks. The model is suitable for simulation-based studies, and the model is made openly available to the community.

## Introduction

Cortical dynamics underlie many cognitive processes and emerge from complex multi-scale interactions. These emerging dynamics can be explored in *large-scale, data-driven, biophysically-detailed* models (*Markram et al., 2015*; *Billeh et al., 2020*), which integrate different levels of organization. The strict biological and spatial context enables the integration of knowledge and theories, the testing and generation of precise hypotheses, and the opportunity to recreate and extend diverse laboratory experiments based on a single model. This approach differs from more abstract models in that it emphasizes *anatomical completeness* of a chosen brain volume rather than implementing a specific hypothesis. Using a 'bottom-up' modelling approach, many detailed constituent models are combined to produce a larger, multi-scale model. To the best possible approximation, such models should explicitly include different cell and synapse types with the same quantities, geometric configuration and connectivity patterns as the biological tissue it represents.

Investigating the multi-scale interactions that shape perception requires a model of multiple cortical subregions with inter-region connectivity, and for certain aspects, the subcellular resolution provided by a morphologically detailed model is also required. In particular, *Barabási et al., 2023* argued that the function of the healthy or diseased brain can only be understood when the true physical nature of neurons is taken into account and no longer simplified into point-neuron networks. Also, *Einevoll et al., 2019* pointed out that simulations of large-scale models are essential for bridging the scales between the neuron and system levels in the brain. In that regard, modern electron-microscopic datasets have reached a scale that allows the reconstruction of a ground truth wiring diagram of local connectivity between several thousand neurons (*Bae et al., 2021*). However, this only covers a small fraction of the inputs a cortical neuron receives. While afferents from outside the reconstructed volume are detected, one can only speculate about the identity of their source neurons and connections between them. The scale required to understand inter-regional interactions is only available at lower resolutions in the form of region-to-region or voxel-to-voxel connectivity data.

To help better understand cortical structure and function, we present a general approach to create morphologically detailed models of multiple interconnected cortical regions based on the geometry of a digitized volumetric brain atlas, with synaptic connectivity predicted from anatomy and biological constraints (*Figure 1*). We used it to build a model of the juvenile rat non-barrel somatosensory (nbS1) regions (*Figure 1*, center). These regions were selected for the wealth of available experimental data from various labs, and to build upon our previous modeling work (*Table 1*). The workflow is based on the work described in *Markram et al., 2015*, with several additions, refinements and new data sources that have been independently described and validated in separate publications (*Table 1*). The model captures the morphological diversity of neurons and their placement in the actual geometry of the modeled regions through the use of voxelized atlas information. We calculated at each point represented in the atlas the distance to and direction towards the cortical surface (*Figure 1*; step 1). We used that information to select from a pool of morphological reconstructions anatomically fitting ones and orient their dendrites and axons appropriately (*Figure 1*; step 2). As a result, the model was anatomically complete in terms of the volume occupied by dendrites in individual layers. We then combined established algorithms for the prediction of local (*Reimann et al., 2015*) and mid-range (*Reimann et al., 2019*) connectivity (*Figure 1*; step 3) as well as extrinsic connectivity from thalamic sources (*Markram et al., 2015*, *Figure 1*; step 4) to generate a connectome at subcellular resolution that combines those scales.

We characterized several emerging aspects of connectivity (*Figure 1*; step 5). First, we found that brain geometry, that is differences in cortical thickness and curvature have surprisingly large effects on how much individual layers contribute to the connections a neuron partakes in. Second, we characterized the predicted structure of connectivity at an unprecedented scale and determine its implications for neuronal function. In particular, we analyzed how the widths of thalamo-cortical axons constrains the types of cortical maps emerging. Furthermore, we characterized the global topology of interacting local and mid-range connectivity, finding highly complex topology of local and mid-range connectivity that specifically requires neuronal morphologies. Finally, we systematically analyzed the higher-order structure of connectivity beyond the level of pairwise statistics, such as connection probabilities. Doing so, we characterized highly connected clusters of neurons, distributed throughout the volume that are tied together by mid-range synaptic paths mediated by neurons in layer 5, which act as 'highway hubs' interconnecting spatially distant neurons in the model. The highly non-random

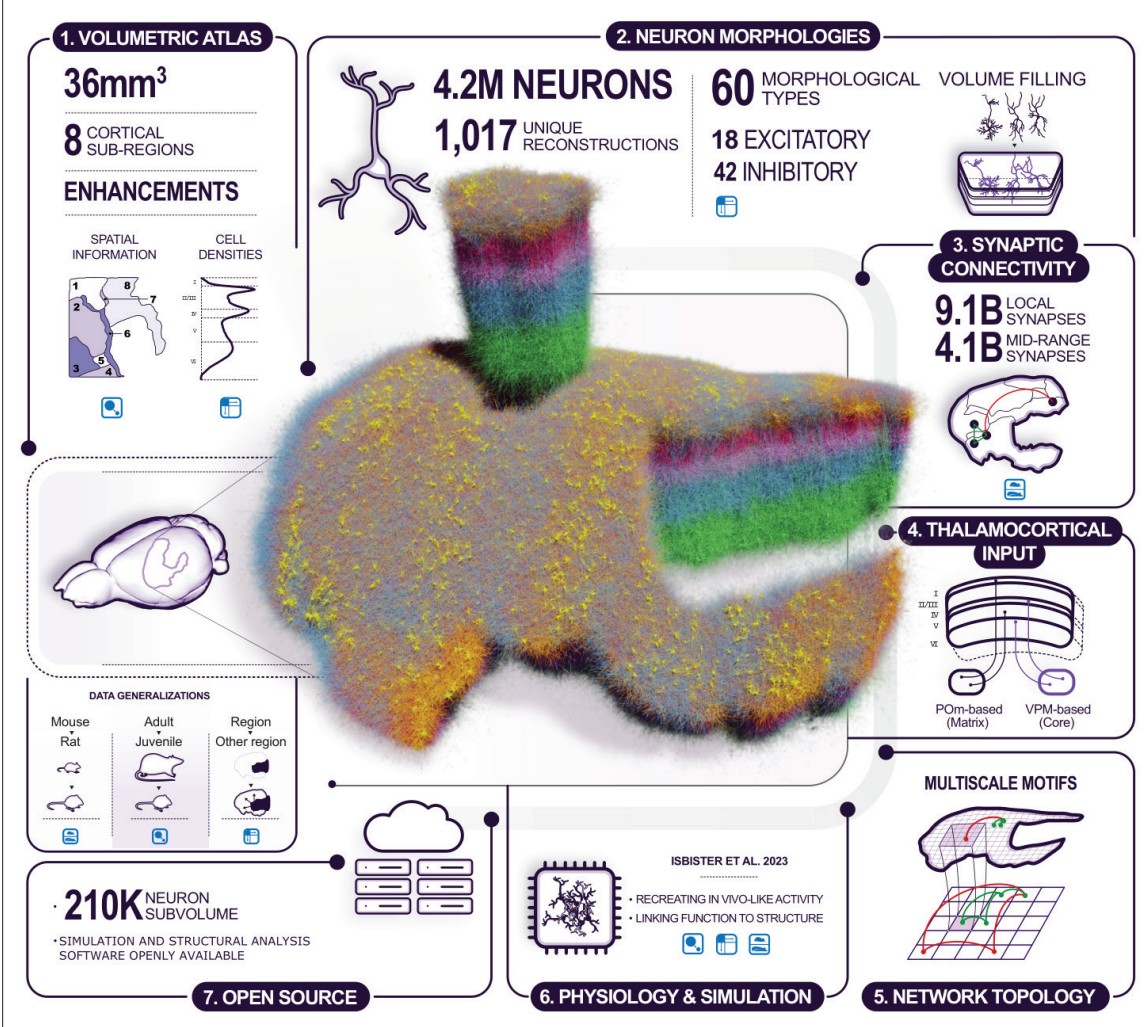

**Figure 1.** Overview of the model building and analysis workflow. Step 1: Building was based on a volumetric atlas of the modeled regions: (1) S1J; (2) S1FL; (3) S1Tr; (4) S1HL; (5) S1Sh; (6) S1DZ; (7) S1DZO; (8) S1ULp. Additional atlases of biological cell densities and local orientation towards the surface were built. Step 2: Neuron morphologies were reconstructed and classified into 60 morphological types (*m-types*). They were placed in the volume according to the densities and orientations from step 1. Step 3: The anatomy of intrinsic synaptic connectivity was derived as the union of one algorithm for *local* connectivity and one for *mid-range* connectivity. Step 4: Extrinsic inputs from two thalamic sources were placed on modeled dendrites according to published methods. Step 5: Taken together, these steps allowed us to predict the topology of connectivity at scale with (sub-)cellular resolution. Step 6: The anatomical model served as the basis of a physiological model, ready to be simulated. This is presented in an accompanying manuscript. Step 7: The model, simulation and analysis tools have been made publicly available. Left: During modeling, three types of generalization had to be made to fulfill input data requirements: from mouse to rat, from adult to juvenile, and from one cortical region to another. Generalizations used are indicated in each step.

The online version of this article includes the following figure supplement(s) for figure 1:

**Figure supplement 1.** Derivation of neuron density depth profiles.

structure of higher-order interactions in the model's connectivity was further validated in a range of follow-up publications using the model (*Ecker et al., 2024b*; *Egas Santander et al., 2024a*; *Reimann et al., 2024*).

Finally, we present an accompanying manuscript that details neuronal and synaptic physiology modeled on top of these results, describes the emergence of an in vivo-like state of simulated activity, and delivers a number of in silico experiments generating insights about the neuronal mechanisms underlying published in vivo and in vitro experiments (*Isbister et al., 2024*; *Figure 1*; step 6).

The anatomically detailed modeling approach provides a one-to-one correspondence between most types of experimental data and the model, allowing the data to be readily integrated. However, this also leads to the difficult challenge to curate the data and decide which anatomical trend should

**Table 1.** References to publications of input data and methods employed for individual modeling steps.

An asterisk next to a reference indicates that substantial adaptations or refinements of the data or methods have been performed that will be explained in this manuscript. In the other cases, a basic summary will be provided and an exhaustive description in the Methods.

**Data**

| Stage | Topic | Reference |
|---|---|---|
| Atlasing | Region annotation atlas | *Paxinos and Watson, 2007* |
| | Orientation, depth and flat map | *Bolaños-Puchet et al., 2023* |
| | Cell density profiles | *Keller et al., 2019* |
| Volume filling | Neuron reconstructions | *Markram et al., 2015* |
| | in vivo neuron reconstructions | *New, original data |
| Synaptic connectivity | Bouton densities and numbers of synapses per connection | *Reimann et al., 2015* |
| | Pathway strengths, synapse density profiles and topographical mapping | *Reimann et al., 2019* |
| Thalamic inputs | Bouton density profiles | *Meyer et al., 2010* |
| | Projection axon lengths and widths | *Economo et al., 2016* |

**Modeling methods**

| Stage | Topic | Reference |
|---|---|---|
| Atlasing | Cell density volume generation | *Keller et al., 2019* |
| Volume filling | Neuron classification | *Kanari et al., 2019* |
| | Neuron placement | *Markram et al., 2015* |
| Synaptic connectivity | Local connectivity | *Reimann et al., 2015* |
| | Inter-region connectivity | *Reimann et al., 2019* |
| Thalamic inputs | Input generation | *Markram et al., 2015* |

be integrated next. Due to the incredible speed of discovery in the field of neuroscience an integrative model will always be lagging behind the latest results, and due to its immense breadth, there is no clear answer to which feature is most important. We believe the solution to this is to provide a validated model with clearly characterized strengths and weaknesses, along with the computational tools to customize it to fit individual projects. We have therefore made not only the model, but also most of our tool chain openly available to the public (*Figure 1*; step 7).

Already the process of adding a new data source to drive a refinement of the model serves to provide important insights. We demonstrate this by comparing the model to connectivity characterized through electron microscopy (*Bae et al., 2021*), finding mismatches, and describing the changes required to fix them. This allowed us to determine which rules are required to predict connectivity from the locations and densities of neuronal processes. Previously, simple overlap of distributions of axonal and dendritic segments has been proposed (Peters' rule; *Peters and Feldman, 1976*; *Garey, 1999*), and contrasted with findings of preference for specific cell types or subcellular domains (*White and Keller, 1987*; *Mishchenko et al., 2010*). Our approach to local connectivity combines overlap with the principle of cooperative synapse formation (*Fares and Stepanyants, 2009*; *Reimann et al., 2015*), additionally *Schneider-Mizell et al., 2024* proposed a combination of overlap and targeting preferences. Our comparison to electron microscopy let us uncover the strength and nature of the targeting preferences shaping connectivity beyond neuronal and regional anatomy. We found that cooperative synapse formation explains some forms of apparent targeting. Additionally, we found

that the distribution of postsynaptic compartments targeted by connections from somatostatin (Sst)-positive neurons is readily predicted from overlap only, while for parvalbumin (PV)-positive and vasoactive intestinal peptide (VIP)-positive neurons additional specificity plays a role. We found no indication of additional specificity for excitatory neurons. The model is available both with and without the characterized inhibitory specificity.

## Results

### Atlas-based neuron placement

The workflow for modeling the anatomy of juvenile rat nbS1 is based on the work described in *Markram et al., 2015*, with several additions and refinements. Most of the individual steps and data sources have already been independently described and validated in separate publications (*Table 1*). The basis of the work was a digital atlas of juvenile rat somatosensory cortex, based on the classic work of *Paxinos and Watson, 2007* (*Figure 1*, step 1.). The atlas provides region annotations, that is each voxel is labeled by the somatosensory subregion it belongs to. It also provided a spatial context for the model, with the non-modeled barrel region being surrounded on three sides by modeled regions (specifically: S1DZ and S1ULp). The atlas was enhanced with spatial data on cortical depth and local orientation towards the cortical surface (*Bolaños-Puchet et al., 2024*; *Bolaños-Puchet et al., 2023*). We further added annotations for cortical layers, placing layer boundaries at fixed normalized depths (*Supplementary file 5*). Using laminar profiles of cell densities (*Figure 1—figure supplement 1*; *Keller et al., 2019*), we generated additional voxelized datasets with cell densities for each morphological type (see Materials and methods).

Next, we filled the modeled volume of the atlas with neurons (*Figure 1*, step 2.). Neuronal morphologies were reconstructed in slices, and repaired algorithmically (*Anwar et al., 2009*; *Markram et al., 2015*). Out of 1017 morphologies, 58 were new in vivo stained reconstructions used for the first time in this work (see Materials and methods). The morphologies were classified into 60 morphological types (*m-types*, see ; 18 excitatory *Figure 2A*, 42 inhibitory), based on expert knowledge and objectively confirmed by topological classification (*Kanari et al., 2019*).

As only 58 morphologies were in vivo stained reconstructions, the rest potentially suffered from slicing artifacts (see *Figure 2—figure supplement 2* for examples), despite applying a repair algorithm (*Markram et al., 2015*). A topological comparison (*Kanari et al., 2018*; *Kanari et al., 2019*) between axons and dendrites of neurons in all layers (*Figure 2—figure supplements 1 and 3*) revealed that in vitro reconstructions from slices could not capture detailed axonal properties beyond 1000 µm, but could faithfully reproduce dendritic arborization.

Cell bodies for all m-types were placed in atlas space according to their prescribed cell densities. At each soma location, a reconstruction of the corresponding m-type was chosen based on the size and shape of its dendritic and axonal trees (*Figure 2—figure supplement 4*). Additionally, it was rotated according to the orientation towards the cortical surface at that point. These steps ensured that manually identified features of the morphologies (see *Supplementary file 6*) landed in the correct layers (*Figure 2C*, *Figure 2—figure supplement 5*). Additionally, a random rotation around an axis orthogonal to layer boundaries was applied.

### Biological dendrite volume fractions emerge in the model

The distribution over 60 m-types in the model captured the great morphological diversity of cortical neurons and matched the reference data used (*Figure 3A*, see Materials and methods). The placement of morphological reconstructions matched expectation, showing an appropriately layered structure with only small parts of neurites leaving the modeled volume (*Figure 2C*). For a more quantitative validation, we calculated the fraction of the volume occupied by neurites in 100 depth bins of a cylindrical volume spanning all layers and with a radius of 100 µm (*Figure 3B*). This included an estimated volume for axons forming the mid-range connectivity between modeled regions, based on its synapse count (see below and Materials and methods).

We found a clearly layered structure also for the neurite volumes, with apical dendrites contributing substantially to the volume in layers above their somas. The volume was clearly dominated by dendrites, filling between 23% and 47% of the space, compared to 2% to 11% for axons (*Figure 3B3*). The range of values for dendrites matched literature closely (between 26% and 46%, *Santuy et al.,*

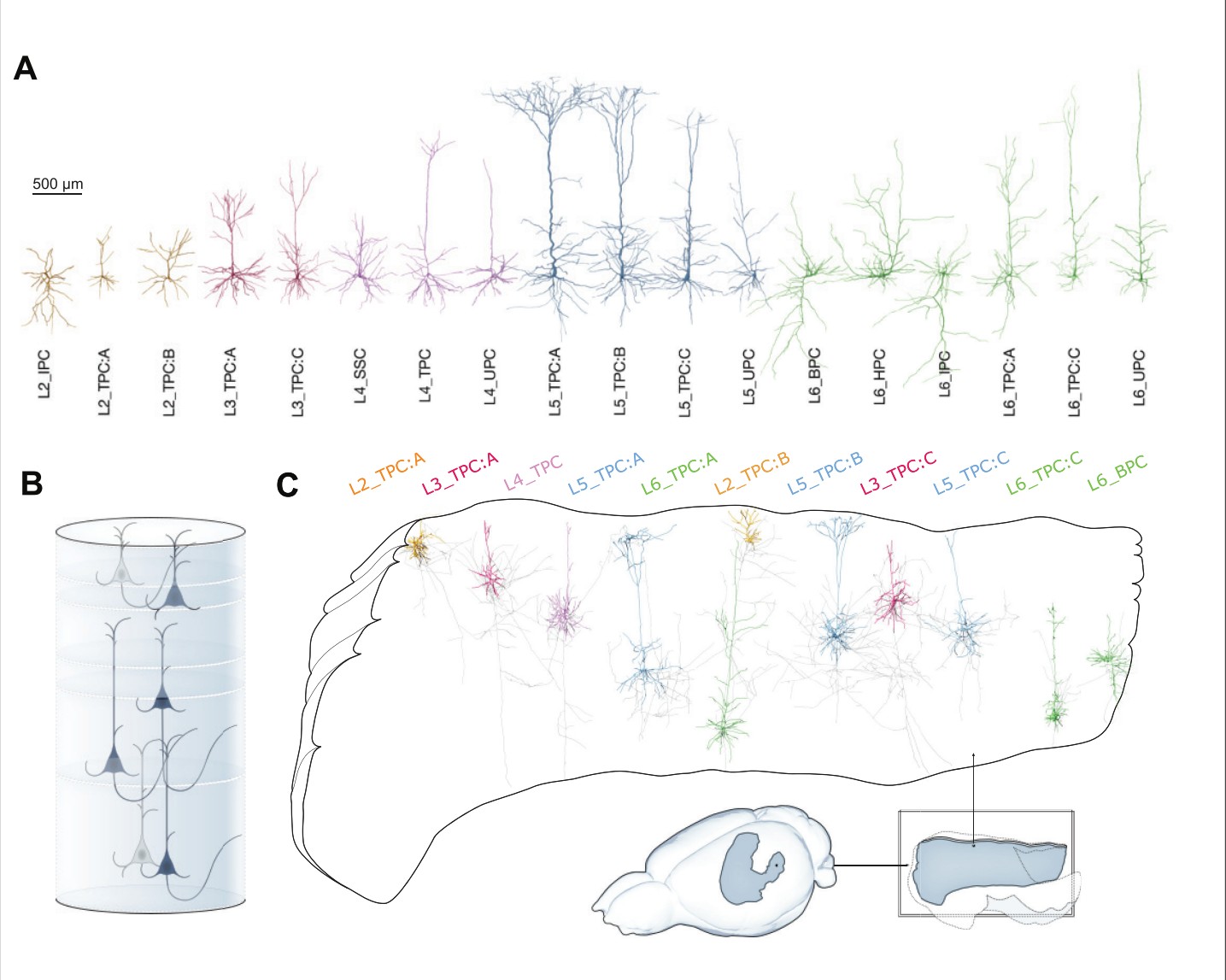

**Figure 2.** Anatomy of the model. (**A**) Exemplar 3D reconstructions of the 18 excitatory m-types. Rendering and visualization was done in NeuroMorphoVis (*Abdellah et al., 2018*). Dendritic diameters are scaled (x3) for better resolution. (**B**) Modeled cortical layers, with exemplars of excitatory morphological types placed in the model. (**C**) The placement of each morphological type recreates the biological laminar anatomy of dendrites and axons, which then serves as the basis of local connectivity. Inset shows modeled brain regions (solid colors) in the context of the non-modeled regions (transparent).

The online version of this article includes the following figure supplement(s) for figure 2:

**Figure supplement 1.** Topological comparison as in *Kanari et al., 2018*; *Kanari et al., 2019* of in vivo stained and in-vitro (i.e. from slices) axonal (**A**) and dendritic (**B**) reconstructions of rat somatosensory cortex of pyramidal cells from layers 2–6.

**Figure supplement 2.** Comparison of representative reconstructructed morphologies that have been stained in slice (top) or in vivo (bottom).

**Figure supplement 3.** Classification and morphometrics of excitatory morphologies.

**Figure supplement 4.** Scoring the placement of a neuron morphology for a voxel.

**Figure supplement 5.** Exemplary neurons in the model, one per m-type, rendered in the context of a slice spanning all cortical layers (grey borders).

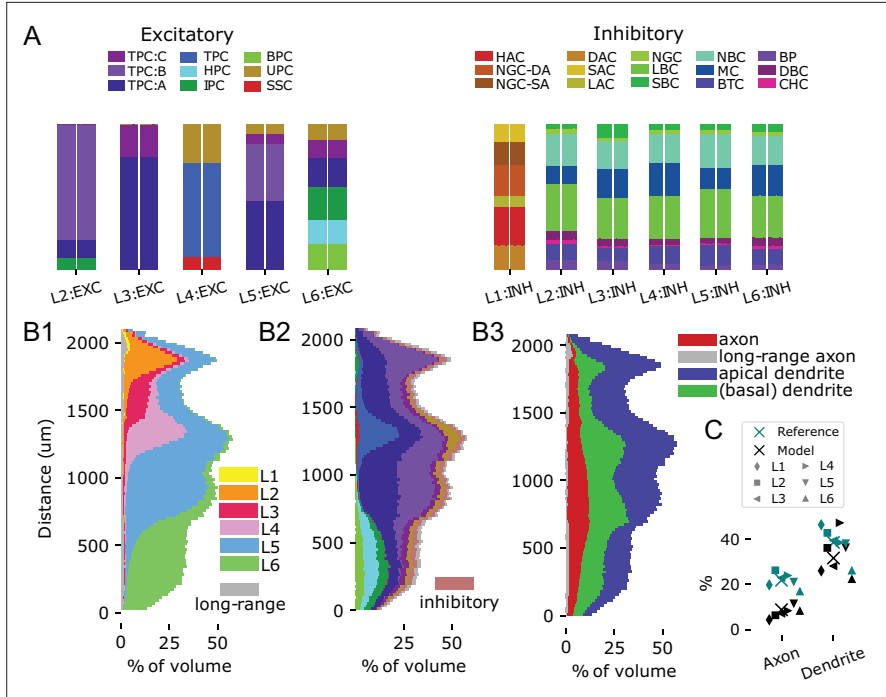

**Figure 3.** Volumetric anatomy of the model. (**A**) M-type composition per layer: For each layer, stacked histograms of the relative fractions of each m-type, comparing the model (right bar for each layer) to the input data (left bar). For simplicity, the layer designation is stripped from each m-type. Left: For excitatory types; right: Inhibitory types. (**B**) Stacked histograms of the fraction of space filled by neurites at various depths in the model. The y-axis indicates the distance in μm from the bottom of layer 6. (**B1**) For neurites of neurons in different layers indicated in different colors. Grey: estimated lower bound for the volume of axons supporting the mid-range connectivity. (**B2**) For neurites of different m-types. Colors as in A, but inhibitory types are grouped together. (**B3**) For different types of neurites. (**C**) Comparing fractions for axons and dendrites to the literature (*Santuy et al., 2018*). X-marks indicate overall means, other marks means in individual layers; teal: reference, black: model.

2018). Conversely, values for axons were lower than literature (between 17% and 24%); however, this is explained by the vast amount of external input axons from non-somatosensory and extracortical sources that are not part of the model.

## Local, mid-range and extrinsic connectivity modeled separately

Synaptic connectivity in comparable models is often modeled by imposing experimentally characterized connection probabilities, potentially distance-dependent, onto the various neuronal pathways (*Potjans and Diesmann, 2014*; *Pronold et al., 2024*). As this approach directly parameterizes structural connections strengths at the population level it allows easy recreation of meso-scale architectures, as characterized in, e.g., *Jiang et al., 2015*. On the other hand, it has been shown that top-down imposed connection probabilities, even distance-dependent ones, cannot fully capture the non-random higher order structure of connectivity at cellular resolution (*Gal et al., 2021*). Such non-random connectivity has been repeatedly demonstrated in cortical circuitry (*Song et al., 2005*; *Perin et al., 2011*; *Reimann et al., 2024*; *Egas Santander et al., 2024a*) and has functional impact (*Reimann et al., 2017b*; *Egas Santander et al., 2024a*; *Nolte et al., 2020*). Instead, we used a previously published approach (*Reimann et al., 2015*) that selects synapses as a subset of axo-dendritic, axo-somatic and axo-axonic appositions, based on biologically motivated rules (*Figure 1*, step 3; 4 A). Note that this algorithm is not equivalent to the so-called 'Peters' rule' (*Peters and Feldman, 1976*; *Garey, 1999*; *Rees et al., 2017*), as the subset is not selected completely randomly. Crucially, the concept of cooperative synapse formation (*Fares and Stepanyants, 2009*) is used, which introduces statistical dependence for potential synapses between the same pair of neurons: If one is selected the probability that others are selected increases. The algorithm has been demonstrated to result in a nonrandom higher-order structure of connectivity matching biological references (*Reimann et al.,*

*2015*; *Gal et al., 2017*; *Reimann et al., 2017a*; *Reimann et al., 2017b*). Conversely, connection probabilities are not directly controlled by the modeler. We measured the emerging connection probabilities by emulating a multi-patch procedure that is often used for that purpose in vitro (*Figure 4B*, *Figure 4—figure supplement 2A*, see Materials and methods). We compared them to connection probabilities reported in 124 literature sources gathered by *Zhang et al., 2019* (*Figure 4—figure supplement 2B–D*). We note that reported connection probabilities vary drastically, for example, ranging from zero to 88% for L2/3 EXC to L2/3 PV. Significant variability persisted when only data sources for rat or the age of the model (P14) were considered, with ranges of values stretching up to 51% and 54%, respectively. While values for the model were sometimes outside the reported range, this was mostly restricted to pathways with only one or two sources.

To create biologically realistic overall neuron out-degrees, our algorithm requires axon reconstructions to be complete. While we try to ensure the completeness of axonal arborizations through the use of in vivo stained reconstructions and repair algorithms, this becomes more challenging, the more an axon stretches away from its soma, due to lack of dye penetration and potentially slicing artifacts. According to our analysis of the morphologies used, most axons were only accurately reconstructed up to 1000 µm from the soma (see above). This contrast between local axon around the soma and more mid-range collaterals stretching into neighboring regions can also be seen in in vivo stained reconstructions (*Figure 4C*). Therefore, to model connections at larger distances, we used a second, previously published algorithm (*Reimann et al., 2019*). It places mid-range synapses according to three biological principles that all need to be separately parameterized based on experimental data (as detailed in the subsequent paragraphs and *Figure 4—figure supplement 1*): First, connection strength: ensuring that the total number of synapses in a region-to-region pathway matches biological estimates. Second, layer profiles: ensuring that the relative number of synapses in different layers matches biological estimates. Third, topographical mapping: Ensuring that the specific locations within a region targeted by mid-range connections of neurons describe a biologically parameterized, topographical mapping. We note that in the literature on macro-scale connectomics, the pathways between somatosensory regions would be considered 'short-range, inter-regional connections'. We will still refer to connections from the two algorithms as the *local* and *mid-range* connectomes respectively, for simplicity and because of the intuitive contrast it invokes.

To parameterize connection strengths, we used data from axon tracing experiments provided by the Allen Institute (*Harris et al., 2019*), adapted from mouse to rat, yielding expected densities of projection synapses between pairs of regions (*Figure 4D1*). First, we asked to what degree the local connectome algorithm suffices to model the connectivity within a region. To that end, we compared the mean excitatory synapse densities of local connectivity to the target values adapted from *Reimann et al., 2019*, diagonal *Figure 4D1* vs. *Figure 4D2*; *Figure 4E*. These target values were derived from relative region-to-region connectivity strengths, estimated by axon tracing, scaled to match overall volumetric synapse densities estimated from electron microscopy. We found that the overall average matches the data fairly well, however the variability across regions was lower in the model ($0.123 \pm 0.017 \mu m^{-3}$; mean ± std in the model vs. $0.097 \pm 0.06 \mu m^{-3}$). Based on these results, we decided that the local connectome sufficed to model connectivity within a region. It also created a number of connections across region borders (*Figure 4D2*). Consequently, we parameterized the strengths of additional mid-range connections to be placed as the difference between the total strength from the data and the strengths resulting from local connectivity, with connection strengths within a region set to zero (*Figure 4G*). As a result synaptic connections between neighboring regions will be placed by both algorithms (*Figure 4F*), with a split ranging from 20% local to 70% local. The lower spread of apposition-based synapse density within a region (see above) will also reduce the variability of combined synapse density from both algorithms (*Figure 4H*). While this will halve the coefficient of variation of density across regions from 0.34 to 0.17, the overall mean density over all regions is largely preserved (0.23 data vs. 0.24 for the combined algorithms; *Figure 4H*, red dot).

Finally, the dendritic locations of synaptic inputs from thalamic sources were modeled as in *Markram et al., 2015*, using experimental data on layer profiles of bouton densities of thalamo-cortical axons, and morphological reconstructions of these types of axons (*Figure 1*, step 4; *Figure 4—figure supplement 4A*; Materials and methods). Based on these data, each thalamic input fiber was assigned an innervated domain in the model, recreating the layer profile along an axis orthogonal to layer boundaries and spreading equally in the other dimensions (*Figure 4—figure supplement 4B–E*). We

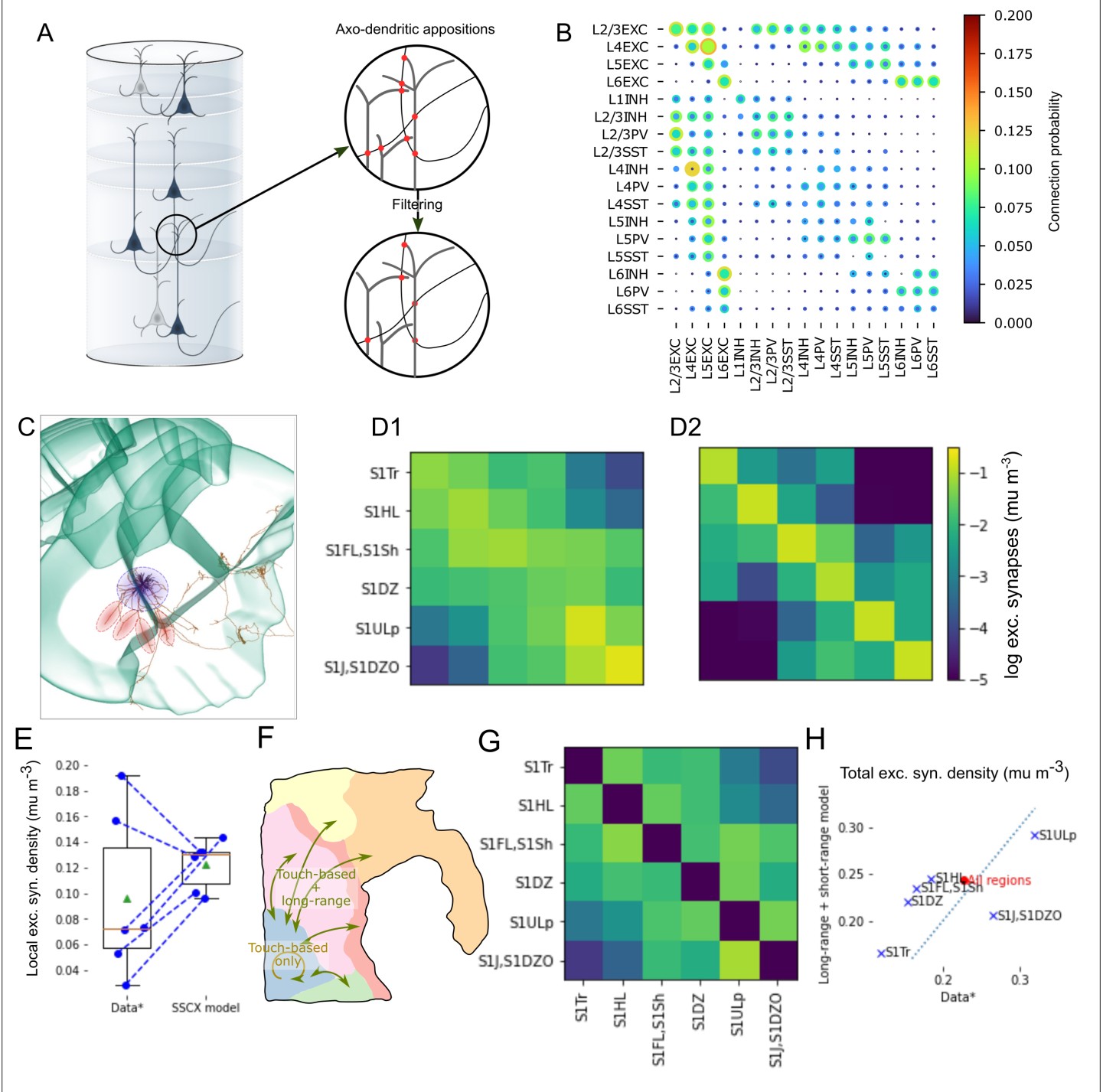

**Figure 4.** Intrinsic connectivity as union of local and mid-range connectivity. (**A**) Local connectivity was derived as in *Markram et al., 2015*; *Reimann et al., 2015*, that is based on neuron placement and morphologies. Right: Axo-dendritic appositions were considered as potential synapses. They were then filtered to yield among other biological constraints biologically realistic bouton densities. (**B**) Range of resulting connection probabilities within.100 µm Indicated are the minimum (inner circle) and maximum (outer circle) found over 20 repetitions of sampling 1000 pairs in a simulated multi-patch procedure (*Figure 4—figure supplement 2A*). See Materials and methods for mapping from morphological types to inhibitory subclasses (PV/SST/INH). (**C**) An exemplary layer 5 PC axon reconstruction from Janelia MouseLight (mouselight.janelia.org, *Gerfen et al., 2018*) that includes mid-range collaterals, shown in the context of mouse somatosensory regions. The blue circle highlights local branches all around the soma. Red highlights depict more targeted collaterals into neighboring regions. (**D1**) Predicted pathway strengths as indicated in *Figure 4—figure supplement 1*. (**D2**) Pathway strength emerging from the application of the apposition-based connectivity algorithm described in *Reimann et al., 2015*. (**E**) Values of the diagonal of (**D1**) compared to the diagonal of (**D2**). (**F**) Schematic of the strategy for connectivity derivation: Within a region only apposition-based connectivity

*Figure 4 continued on next page*

*Figure 4 continued*

is used; across regions the union of apposition-based and mid-range connectivity. (**G**) Connection strength constraints for the mid-range connectivity derived by subtracting D2 from D1 and setting elements < 0 to 0 (colors as in D). (**H**) Resulting total density from both types of modeled synaptic connections in individual regions compared to the data in (**D1**).

The online version of this article includes the following figure supplement(s) for figure 4:

**Figure supplement 1.** Data sources for connectivity modeling.

**Figure supplement 2.** Comparing local connection probabilities to the literature.

**Figure supplement 3.** Validation of modeled connectivity.

**Figure supplement 4.** Modeling thalamo-cortical synaptic inputs.

modeled two types of thalamic inputs, based on the inputs into barrel cortex from the ventral postero-medial nucleus (VPM-based) and from the posterior medial nucleus (POm-based), respectively (**Harris et al., 2019**; **Shepherd and Yamawaki, 2021**). While barrel cortex was not a part of the model, we used these projections as examples of a *core*-type projection, providing feed-forward sensory input (VPM-based) and a *matrix*-type projection, providing higher-order information (POm-based).

The numbers of thalamic input fibers innervating the model were estimated as follows. Laminar synapse density profiles were summed over the volume to estimate the total number of thalamo-cortical synapses, and the number of synapses per neuron was estimated from the lengths of thalamo-cortical axons. The ratio of these numbers resulted in 72,950 fibers for the POm-based matrix-type projection, and 100,000 fibers for the VPM-based core-type projection. These numbers are consistent with the volume ratio of the two thalamic nuclei ($1.25mm^3$ for POm to $1.64mm^3$ for VPM; ratio: 0.76).

## Specificity of axonal targeting

For the axons of various inhibitory neuron types, certain aspects of connection specificity have been characterized (**Tremblay et al., 2016**), such as a preference for peri-somatic innervation for PV+ basket cells. The local connectivity in the model is based axo-dendritic overlap, combined with a pruning rule that prefers multi-synaptic connections, but does not take the post-synaptic compartment type into account (**Reimann et al., 2015**). As such, it captures targeting preferences resulting from specific axonal morphologies, but not potential effects due to potential molecular mechanisms during synapse formation or pruning. This is based on the idea that the developmental mechanisms underlying connectivity are manifold and complicated, but their cumulative effect is at least partially visible in the characteristic shapes of the neurites. It has been demonstrated by us and other groups that anatomy-based approaches suffice to create highly nonrandom network topologies that match many biological trends (**Gal et al., 2017**; **Reimann et al., 2017a**; **Udvary et al., 2022**), but the analyses have so far been focused mainly on excitatory sub-networks. To investigate to what degree the approach suffices to explain observed targeting trends, we compared the model to characterizations of targeting specificity from the literature, recreating the electron microscopy studies of **Motta et al., 2019** and **Schneider-Mizell et al., 2024** in silico.

**Motta et al., 2019** analyzed an approximately 500,000 μm³ volume of tissue in layer 4, considering the postsynaptic targets of the contained axons. As axons in the volume were fragments, specificity was assessed by comparing the data to a binomial control fit against observations of axons forming at least one synapse onto a postsynaptic compartment type (see Materials and methods). Axons with unexpectedly high fractions onto a compartment type were then considered to target that type. We calculated targeting fractions of axons forming at least 10 synapses in a comparable layer 4 volume of our model (*Figure 5A*). Compared to **Motta et al., 2019**, fractions of synapses onto apical dendrites were elevated, and more inhibitory synapses were considered (*Figure 5B*). As in the original study, for almost all compartment types, observed distributions were more long tailed than expected with a number of axons showing a significantly high targeting fraction (*Figure 5C*). The fractions of axons with such specificities also matched the reference, except for an even higher fraction being specific for apical dendrites in the model (*Figure 5D*). The match resulted from local connectivity in our model using the principle of *cooperative synapse formation* (**Fares and Stepanyants, 2009**), where all synapses forming a connection are kept or pruned together. This creates a statistical dependence between the synapses that results in a significant deviation from the binomial control models. We demonstrate this with a simple stochastic model of cooperative synapse formation, fitted to the data

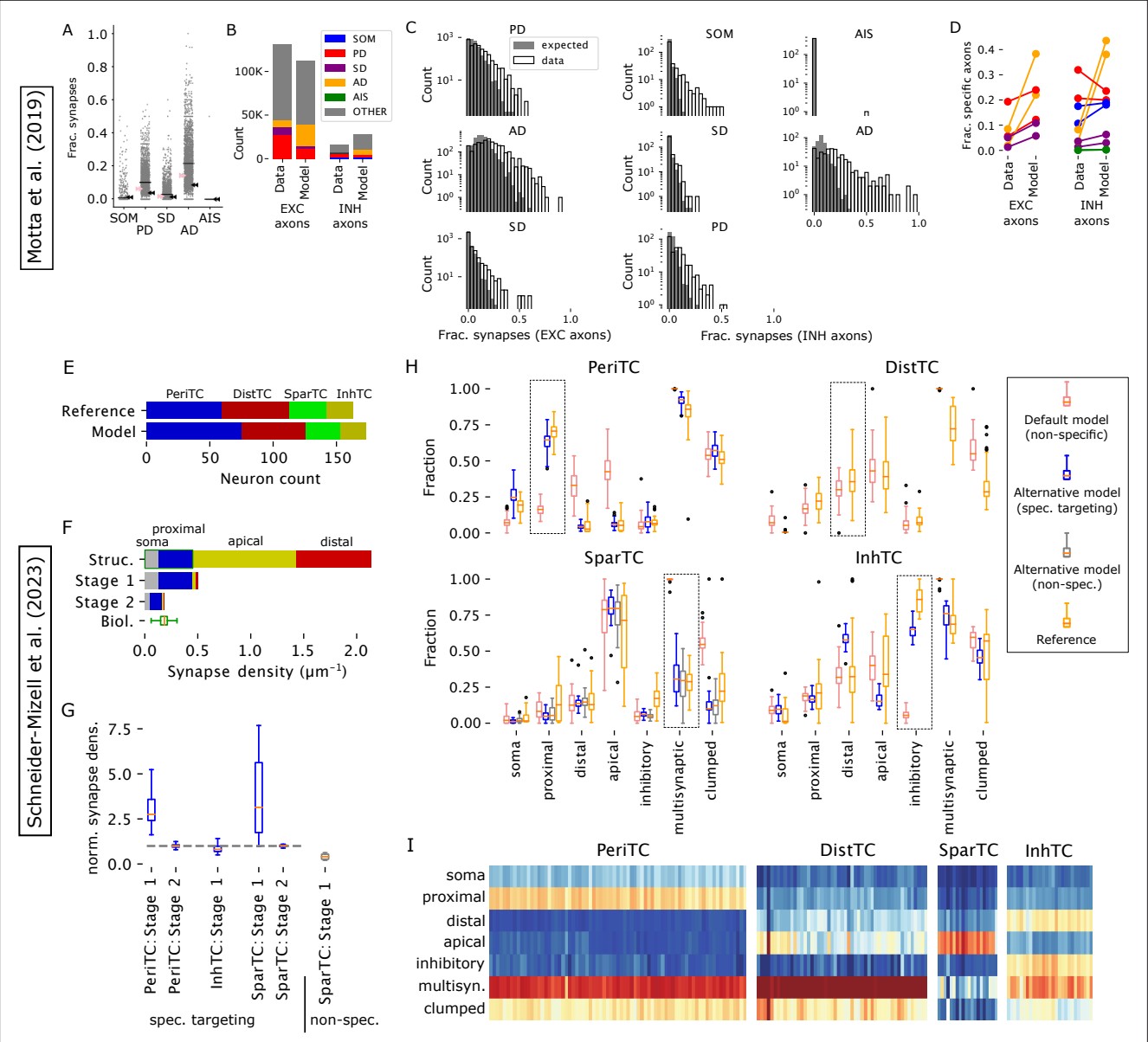

**Figure 5.** Recreating an electron-microscopic specificity analysis In the top part we analyzed synapses, axons and dendrites in an approximately cubic volume in L4, emulating the techniques of *Motta et al., 2019*. In the bottom part we analyzed a 100 x 100 µm volume of the MICrONS dataset defined by *Schneider-Mizell et al., 2024*. (**A**) Fraction of synapses placed on somata (SOM), proximal dendrites (PD), smooth dendrites (SD), apical dendrites (AD) and axon initial segments (AIS) for axon fragments in the sampled volume. Black bars indicate mean values over axons, arrows indicate binomial probabilities fit to observations of axons forming at least a single synapse; pink: for excitatory axons, black: inhibitory. Fractions missing from 100% are onto other compartment types. (**B**) Overall count on synapses on different postsynaptic compartment classes inside the studies volumes, comparing the data of *Motta et al., 2019* to the model. Colors as indicated in the legend. (**C**) Distributions of the fractions of synapses onto different compartment types over excitatory (left) and inhibitory (right) axons. Expected from the binomial model in A (grey) against the observations in our anatomical model (black outlines). (**D**) Fraction of axons with significantly increased synapse counts onto different compartment types compared to the binomial control. Indicated for two values of the false detection rate criterion (q=0.05, 0.3, *Storey and Tibshirani, 2003*). Comparing the reference data to our anatomical model. (**E**) Top numbers of neurons in four connectivity-derived classes in the 100x100 µm volume of the MICrONS dataset, defined by *Schneider-Mizell et al., 2024*. Bottom: Numbers in a comparable volume of the model. For assignment of m-types into the four classes see the main text. (**F**) Derivation of alternative connectivity with targeting specificity, using the example of the 'PeriTC' class. The axo-dendritic appositions of an axon (top) are classified as matching the targeting (green box; here: appositions with somata or proximal dendrites) or not. Non-matching appositions are removed with probability $p_{nt}$ (middle). This is followed by a non-specific removal of connections (formed by single or multiple synapses) until the biological density of synapses on the axon is met (bottom). (**G**) Bouton densities resulting from the process in F. Left to right: For PeriTC neurons, targeting somata and proximal dendrites; for InhTC neurons, targeting inhibitory neurons; for SparTC neurons, targeting the first synapse of a connections; for SparTC

*Figure 5 continued on next page*

*Figure 5 continued*

neurons, without targeting preference. Respective optimized $p_{nt}$ values reported in the main text. (**H**) Resulting targeting of postsynaptic compartments, fraction of synapses in multisynaptic connections and clumped synapses (see *Schneider-Mizell et al., 2024*). Red: Default model; blue: optimized alternative model with specific targeting; grey: optimized alternative model without specific targeting; orange: data of *Schneider-Mizell et al., 2024* Boxes outline the characteristic feature of each class. (**I**) Inhibitory targeting specificities combining the best performing models.

The online version of this article includes the following figure supplement(s) for figure 5:

**Figure supplement 1.** Stochastic control of targeting.

---

of *Motta et al., 2019*. For each axon in the data, we generated a random list of compartments types for potential synapses matching the overall frequencies observed in the data. For each, the number generated was 2.5 times the number of synapses of the corresponding axon. Next, we pruned synapses to the original counts in two ways: First, we grouped potential synapses onto the same type into groups of three that are kept or discarded together, thereby introducing statistical dependence. The results provide the same amount of apparent targeting as the data (*Figure 5—figure supplement 1A-C*). Second, we pruned without the grouping where no targeting was found (*Figure 5—figure supplement 1D-F*). We do not claim that the stochastic model and its parameters provide an accurate description of the underlying biological processes, only that it demonstrates the effect of cooperative synapse formation. We conclude that the non-random trends observed by *Motta et al., 2019* are an indication of cooperative synapse formation captured by our model.

Recently, the MICrONS dataset (*Bae et al., 2021*) has been analyzed with respect to the axonal targeting of inhibitory subtypes in a subvolume of 100 x 100 µm$^2$ surface area spanning all layers of the cortex (*Schneider-Mizell et al., 2024*). Similar to *Motta et al., 2019* they considered their distributions of types of postsynaptic compartments. But due to the larger reconstructed volume, they were able to analyze complete or almost complete axons, allowing for a quantitative comparison rather than relying on a fitted binomial control model. Additionally, they calculated the fraction of synapses that are part of a multisynaptic connections and the fraction thereof that is within 15 µm of another synapse of the same connection.

A comparable volume of the model (see Materials and methods) contained 173 interneurons vs. 163 in the original study. Their distribution into four connectivity classes according to morphological and molecular determinants hypothesized by *Schneider-Mizell et al., 2024* approximately matched as well (*Figure 5E*; perisomatic targeting: Basket Cells; distal targeting: Sst+; sparsely targeting: Neurogliaform Cells and L1; inhibitory targeting: bipolar and VIP+). The postsynaptically targeted compartments matched for the distal targeting group (*Figure 5H*, top right; *Figure 5I*), indicating that their preference can be explained by their axonal morphology, specifically its trend to ascend to and then branch in superficial layers. For the other three classes, their respective eponymous trends were not recreated (*Figure 5H*, red vs. orange).

We then explored the strengths of targeting mechanisms required to explain postsynaptic targets in a modified version of our connectivity algorithm: Beginning with all axo-dendritic appositions as potential synapses, we first remove appositions that are not placed on the preferred postsynaptic compartment with a probability $p_{nt}$ (*Figure 5F*; top to middle). This replaces a non-specific, but otherwise identical first pruning steps in the regular version of our algorithm. This is followed by removing connections non-specifically, until the biological density of synapses on the axon is matched (*Figure 5F*; middle to bottom). As reference for biological densities of synapses on axons, we use the sources listed in *Reimann et al., 2015*.

For the perisomatic targeting class, probability $p_{nt}$ to remove non-proximal, non-soma synapses was optimized against the data of *Schneider-Mizell et al., 2024* to 97%. The remaining synapses had a density on the axons of perisomatic targeting cells that was three times higher than biology, which could be reduced in the second, non-specific pruning step removing two thirds of the connections (*Figure 5G*, left). This indicates substantial room for rewiring through structural plasticity while preserving the targeting specificity of perisomatic targeting cells. The resulting specificity of postsynaptic compartments and multi-synaptic connections then match the reference data (*Figure 5H*, top left, blue vs orange; *Figure 5I*).

For the inhibitory targeting class, probability $p_{nt}$ to remove synapses on non-inhibitory neurons was optimized to a similar value of 96.5%. Curiously, this first step already reduced the resulting axonal

density of synapses to the biological value, indicating that this class of interneurons cannot perform substantial rewiring without losing its targeting specificity (*Figure 5G*, second from left).

For sparsely targeting cells, we evaluated two hypotheses: First, we note that this targeting class is associated with Neurogliaform Cells, which are known to have volumetrically transmitting synapses. It is possible that the sparseness of their targeting can be explained by very few of their synapses having an anatomical postsynaptic partner, rather than by a targeting mechanism. Indeed a non-specific removal of 96% of all synapses recreated the sparsity of these connections found in *Schneider-Mizell et al., 2024* (*Figure 5H*, bottom left, grey vs orange; *Figure 5I*). This reduced the axonal density of synapses to 30% of the biological value, implying that the remaining 70% may be volumetrically transmitting (*Figure 5G*, right). Second, we randomly picked a 'first' synapse from each connection formed by this class that we considered to be targeted. Of the remaining synapses, we removed $p_{nt}$ = 95%, which recreated the characteristic sparsity of the connections equally well (*Figure 5H*, bottom left, blue vs orange; *Figure 5I*). In this case, the second, non-specific pruning step was required, indicating substantial room for rewiring (*Figure 5G*, second from right).

Using a recently developed computational tool ('Connectome-Manipulator'; *Pokorny et al., 2024a*), we created a rewired connectome instance that combines the existing excitatory connectivity with the additional rules for the inhibitory connectivity in a single connectome. This rewired connectome is openly available in SONATA format (*Dai et al., 2020*), see key resources table in the Materials and methods.

## Structure of thalamic inputs

Though we have found that the anatomy-based prediction of connectivity underestimates the specificity of some inhibitory connection types, it remains a powerful tool that has been demonstrated to recreate non-random trends of excitatory connections that make up the majority of synapses (*Reimann et al., 2015*; *Reimann et al., 2017b*; *Gal et al., 2017*). We therefore set out to characterize the anatomy and topology of connections at all scales considered in the model. We began with the thalamic input connections, whose axons were generated based on the axon density profiles and the statistics of their horizontal spread (*Figure 6—figure supplement 1*). Synapses were established based on their volumetric overlap with cortical dendrites to match the laminar synapse density profiles of the presynaptic thalamic axons (see Materials and methods and *Figure 4—figure supplement 4*). The individual postsynaptic dendritic morphologies, cell placement, and orientation were considered in this process. The process did not prefer the dendrites of one neuron type over another. In contrast, *Cruikshank et al., 2007*; *Cruikshank et al., 2010*; *Sermet et al., 2019* found physiologically stronger thalamo-cortical inputs onto excitatory and PV-positive neurons than onto SST-positive neurons. While the authors of the studies make compelling arguments that this at least partially reflects the anatomical strength of the pathways, others have cautioned against mixing of the anatomy and physiology of connectivity (*Sporns, 2013*). We therefore believe that further validation of the thalamo-cortical pathways is best performed in the accompanying manuscript (*Isbister et al., 2024*), where the experimental methods of the references are closely recreated.

Regarding the laminar structure, we found for both projections that the peaks of the mean number of thalamic inputs per neuron occur at lower depths than the peaks of the synaptic density profiles (*Figure 6A*). This is consistent with synapses on apical dendrites of PCs being higher than their somas, but the fact that most peaks occur at places where the synapse density is close to zero gives a clear indication that synapse density profiles alone can be misleading about the location of innervated neurons. At the level of individual neurons, the number of thalamic inputs varied greatly, even within the same layer (*Figure 6B*). Overall, the matrix-type projection innervated neurons in layers 1 and 2 more strongly than the core-type projection, while in layers 3, 4, and 6, the roles were reversed. Neurons in layer 5 were innervated on average equally strongly by both projections, although layer 5a preferred the matrix-type and layer 5b the core-type projection.

To characterize the horizontal structure, we introduced the common thalamic innervation (CTI, see Materials and methods) as a measure of the overlap in the thalamic inputs of pairs of neurons. As pairs with many common inputs are likely to have similar stimulus preferences, the magnitude and range of this effect has consequences for the emergence of functional assemblies of neurons. As expected from the horizontal extent of individual fibers, the CTI was distance-dependent (*Figure 6C,D*), and showed strong variability. Even directly neighboring pairs might not share a single thalamic afferent, leading to a sparse

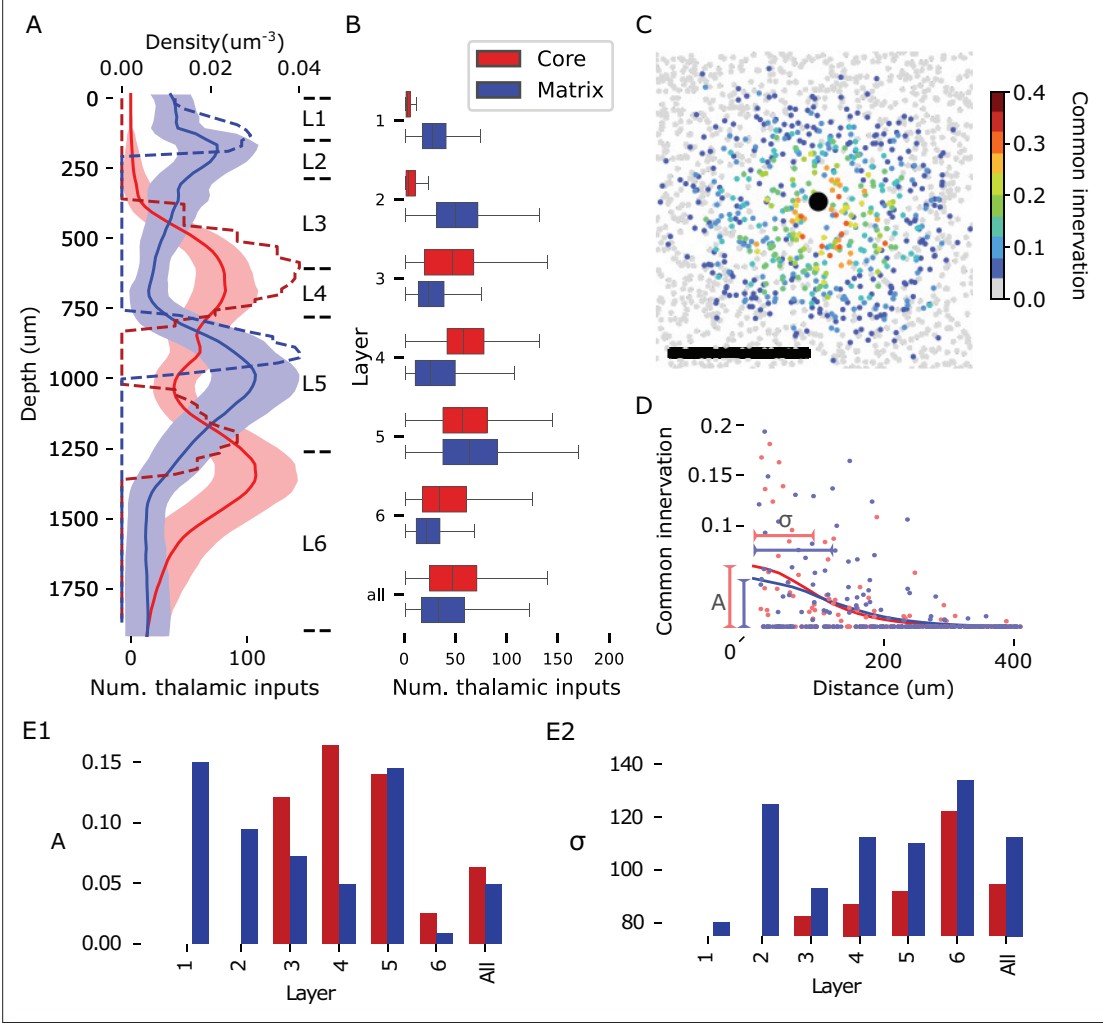

**Figure 6.** Anatomy of thalamic innervation. (**A**) Depth profiles of synapse densities (dashed lines) and mean number of thalamic inputs per neuron (solid lines) for core- (red) and matrix-type (blue) thalamo-cortical projections. Shaded area indicates the standard error of mean. For a region-specific validation of synapse densities (dashed lines) against experimental data, see *Figure 6—figure supplement 2*. (**B**) Mean and standard deviation of the number of thalamic inputs for neurons in individual layers or all neurons. Colors as in A. (**C**) Common thalamic innervation (CTI) of an exemplary neuron (black dot) and neurons surrounding it, calculated as the intersection over union of the sets of thalamic fibers innervating each of them. Scale bar: 200 μm. (**D**) CTI of pairs of neurons at various horizontal distances. Dots indicate values for 125 randomly picked pairs; lines indicate a sliding average with a window size of 40 μm. We perform a Gaussian fit to the data, extracting the amplitude at 0 μm (**A**) and the standard deviation (σ). (**E**) Values of A (unitless) and σ (in flatmap coordinates) for pairs in the individual layers or all pairs. Colors as in A.

The online version of this article includes the following figure supplement(s) for figure 6:

**Figure supplement 1.** Input data for the derivation of the locations of thalamic inputs.

**Figure supplement 2.** Validation of thalamo-cortical density profiles.

spatial distribution of pairs with strong overlap. A Gaussian fit of the distance dependence of CTI revealed roughly equally strong overlapping innervation for both core- and matrix-type projections (*Figure 6E1*). The strength of the overlap increased for lower layers in the case of core-type and decreased for matrix-type projections, while being equally strong in layer 5. The horizontal range of common innervation was larger for matrix-type projections in all layers (*Figure 6E2*). In summary, while the anatomy of thalamo-cortical projections introduces a spatial bias into the emergence of cortical maps, it is relatively weak on its own and supports different stimulus preferences even for neighboring pairs of neurons.

## Local brain geometry affects connectivity

Cortex is often seen as a homogeneous structure with parallel layers, but at the larger spatial scales considered in this model, significant variability in its height and curvature can be observed. Our

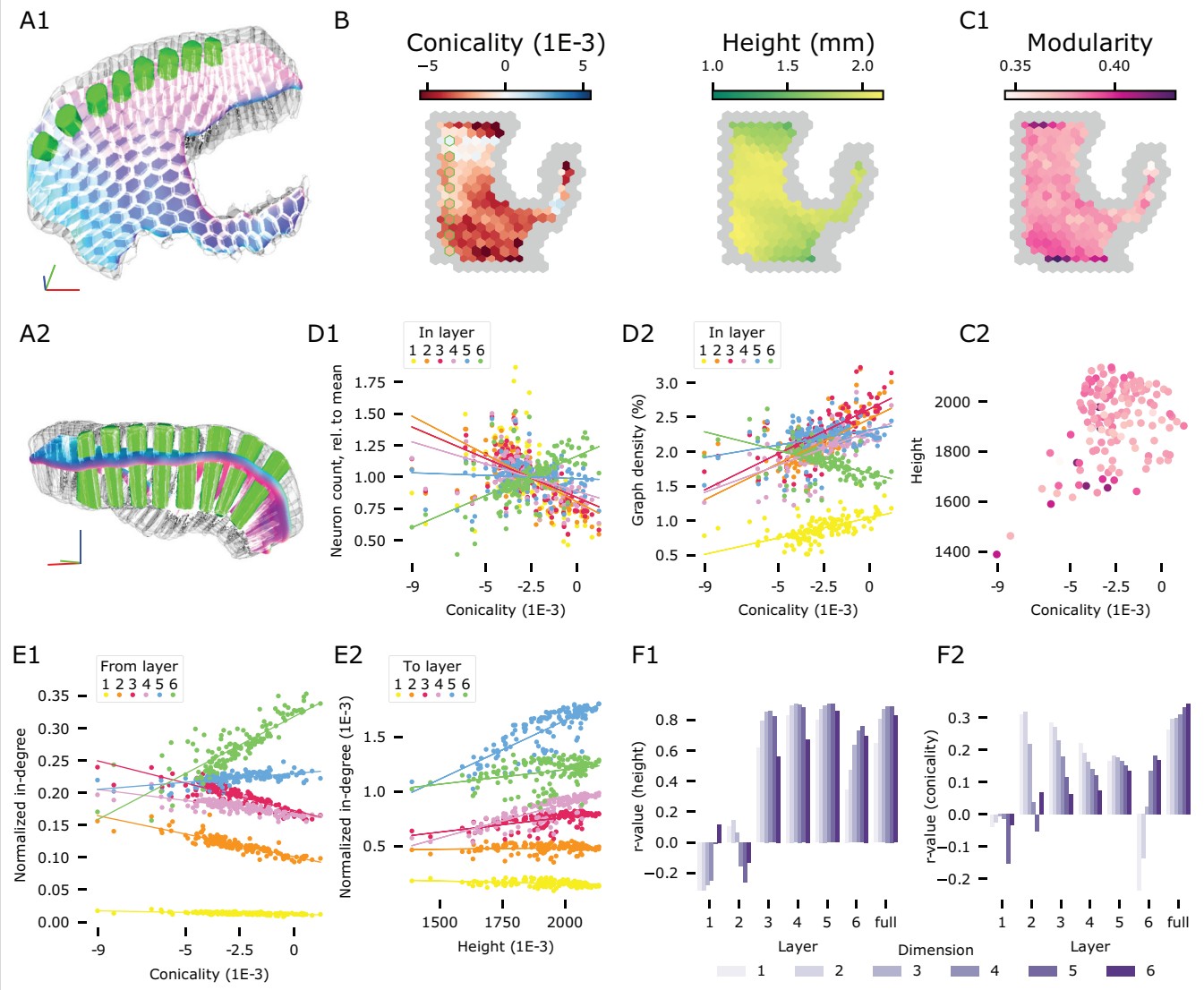

**Figure 7.** A1, A2: Parcellation of the modeled volume into 230 µm radius columns. Exemplary slice of columns highlighted in green. (**B**) Geometrical metrics of column subvolumes in the flat view. Peripheral columns masked out (grey); green outline: highlighted columns in A. Left: Conicality, defined as the slope of a linear fit of depth against column radius. Negative values indicate narrowing towards L6. Right: Column height, i.e. cortical thickness at the location of the column. (**C1**) Modularity of the networks of connections within each column. (**C2**) Column conicality and height colored by modularity; (**D1**) Conicality of columns against their laminar neuronal composition, normalized against the overall composition of the model. Colored lines indicate linear fits. (**D2**) Conicality against the density of connections in subnetworks given by the intersections of columns with individual layers. (**E**) Counts of afferents formed onto neurons in individual columns from neurons in the entire model. (**E1**) Normalized in-degrees originating from neurons in individual layers, plotted against conicality. (**E2**) In-degree of neurons in individual layers normalized by the overall in-degree in the model into each layer plotted against column height. (**F**) r-values of linear fits against generalized, n-dimensional in-degree as in E2. (**F1**) Of generalized in-degree against height; (**F2**) Of generalized in-degree against conicality.

The online version of this article includes the following figure supplement(s) for figure 7:

**Figure supplement 1.** Deviance and diversity of the morphological composition.

**Figure supplement 2.** Modularity across sub-columns.

approach to connectivity allowed us to predict the impact of local brain geometry on connectivity (see also *Ronan et al., 2011*). We partitioned the model into columnar subvolumes with radii of approximately 230 µm (*Figure 7A*) that we then analyzed separately. We do not claim that these columns have a biological meaning on their own, they only serve to discretize the model into individual data points each representing localized circuitry affected by local geometry.

We quantified the two main variable factors of geometry, measuring the height and *conicality* (a feature that measures the convexity of a column, see Materials and methods) of each column (*Figure 7B*). We found that differences in these factors lead to differences in relevant topological parameters, such as the modularity of the local network (*Figure 7C*). This effect was mostly mediated by differences in the neuronal composition, both in terms of total neuron count (not shown) and relative counts for individual layers (*Figure 7D1*). However, conicality also affected the density of connections in local, layer-specific subnetworks (*Figure 7D2*), that is, a measure that is normalized against neuron counts.

Modularity in particular, is strongly correlated with column volume and neuronal count (*Figure 7—figure supplement 2A,B*), which are in turn driven by the column's height and conicality. This effect is shaped by the network structure and not just by its size, since it is not present for controls where the number of neurons and their connections are maintained but their pairing is assigned at random (i.e. ER-controls). In fact, for ER-controls, modularity is anti-correlated with volume and neuron count (*Figure 7—figure supplement 2B*). One way in which the results could be artificial is as follows: Modularity is first defined for a partition of neurons into modules; next the partition maximizing the measure is considered. This can only be done heuristically, and it is possible that optimal partitions are more readily found in certain connectomes than others (see also Materials and methods). To rule this out, we focused on the most modular columns (i.e. those with modularity higher than 0.4). For these, we showed that on average random subnetworks on 50% of the neurons maintained the modularity values of the full columns and that this is not an artifact of the algorithm computing the modularity values (*Figure 7—figure supplement 2C*).

Going beyond the local, purely internal networks, we considered how much each column was innervated by the individual layers of the entire model. We predict a surprisingly strong impact of geometry, for example the ratio of inputs from layer 3 to inputs from layer 6 shifts from almost 2–1 in convex regions to 1–2 in concave regions (*Figure 7E1*). On the other hand, we consider the total amount of inputs received, in each layer of a column. In layers 4, 5, and 6, neurons had a higher in-degree if they were members of tall columns (i.e. placed at locations of large cortical thickness). For layers 1, 2, and 3 the trend was weakened or nonexistent (*Figure 7E2*). The notion of in-degree can be generalized to the *n-dimensional in-degree* measuring participation in specific, directed motifs of n+1 neurons (see Materials and methods). While trends differed between individual layers, overall the dependence of generalized in-degree on geometrical measures increased with dimension (*Figure 7F1 and F2* 'full'). This was particularly driven by neurons in layer 6. Curiously, in that layer the sign of the r-values, with respect to conicality, inverted from dimensions 1 and 2 to dimensions above 2, indicating the overall innervation of layer 6 is stronger in convex regions, but the participation in higher-order motifs is stronger in concave regions.

The effect of conicality can be explained by convex / concave regions allocating more or less relative space to individual layers, thereby weighing the contributions of the corresponding subnetworks differently. The effect of column height is twofold: First, taller columns will contain more neurons. Second, a different selection of neuron morphologies will be placed, depending on the column height. Briefly, taller morphologies will be selected for taller columns (see also *Figure 2—figure supplement 4*). Indeed, the selected morphologies in layers 2–5 were significantly different from the overall composition in columns taller than 2 mm or shorter than 1.4 mm; in layer 6 only the short columns led to significant deviance (*Figure 7B*, *Figure 7—figure supplement 1A*). Due to the finite number of morphology reconstructions used there is the risk of an artificial effect: It is possible that in geometrically outlying columns only a small number of non-representative morphologies were placed. Testing this, we found that the morphological diversity overall increased from superficial to deeper layers, as more morphological reconstructions were available for the thicker, deeper layers (*Figure 7—figure supplement 1B*). Over columns, diversity was relatively uniform. The largest decrease in diversity was observed for short columns and layer 5. In that layer, the least diverse column used around a third of the available reconstructed morphologies (*Figure 7—figure supplement 1C*). As this still amounted to 909 different morphological reconstruction with significant spread between them, we do not believe that dominance by non-representative morphologies explains the qualitative differences in connectivity.

## The complexity of local and mid-range connectivity requires neuronal morphologies

So far, we have analyzed single (thalamo-cortical) pathways making up less than 5% of the synapses in the model, and connectivity at scales that are already readily achievable in electron-microscopic

reconstructions. The large size of the model also allowed us to characterize predicted connectivity at cellular resolution, but scales that are experimentally only accessible with regional or voxelized resolution (*Oh et al., 2014*; *Bota et al., 2015*; *Scannell et al., 1995*; *Scholtens et al., 2014*), thereby bridging the scales as outlined in the introduction. Note that the topological methods in the following sections represent and analyze the connectome as a graph, with neurons as nodes and connections between neurons as directed edges, irrespective of the number of synapses in the connection.

We began by analyzing the global structure of neuron-to-neuron connectivity in the entire model, considering local and mid-range connectivity separately. The topology of synaptic connectivity at single neuron resolution has previously been described in terms of the over-expression of directed simplices (*Reimann et al., 2017b*). A *directed simplex of dimension n* (or *n*-simplex, plural *n*-simplices) is a neuron motif of $n + 1$ neurons that are connected in a purely feed-forward fashion, with a single *source* neuron sending connections to all others, a single *sink* neuron receiving connections from all others, and the connections between non-sink neurons forming an $n - 1$-simplex (*Figure 8A1* inset; *Figure 8—figure supplement 2A*). In particular, 0-simplices correspond to single nodes, and 1-simplices to directed edges. Simplex counts of different dimensions in a network provide a metric of network complexity and can be used to discern their underlying structure (*Kahle, 2009*; *Curto et al., 2013*; *Giusti et al., 2015*). Regarding function, high-dimensional simplices have been demonstrated to shape the structure of spiking correlation between neurons and membership in functional cell assemblies (*Reimann et al., 2017b*; *Ecker et al., 2024a*).

In line with previous results (*Reimann et al., 2017b*), we found simplices up to dimension seven in the local connectivity (*Figure 8A1*, green). The maximal dimension did not increase compared to *Markram et al., 2015* even with the larger scale of the present model, in accordance with using the same algorithm for local connectivity. However, the addition of mid-range connectivity did produce a major change. In mid-range connectivity alone, simplices of dimension up to 15 were observed (*Figure 8A2*, orange). This holds true even though local and mid-range connectivity have roughly the same number of edges (2.1 billion local, 2.5 billion mid-range), indicating that the higher simplex counts are not simply due to a larger number of connections. The simplex counts in the combined network of local and mid-range connections are not simply the sum of the local and mid-range simplex counts (*Figure 8A2*). They are consistently higher and also attain a higher dimensionality generating motifs of up to dimension 18, indicating a strong structural link between the two systems.

We compared the simplex counts of the model to a range of relevant controls that capture simple anatomical properties, such as the density of connections or cortical layers, but ignore the impact of neuronal morphologies (*Figure 8A1,A2*). This allowed us to assess the degree to which the neuronal geometry generates the complexity of the network. The control models and the parameters on which they capture were the following: the *Erdős–Rényi (ER)* model used the overall connection density, the *stochastic block model (SBM)* used density in m-type-specific pathways, the *configuration model (CM)* used sequences of in- and out-degrees, and finally the *distance block model (DBM)* used distance-dependence and neuron locations for individual m-type-specific pathways. See Materials and methods for details.

We found that the structure of connectivity is not only determined by the parameters captured by the controls. The CM control was the closest control for mid-range connectivity, suggesting that the effect of degree is important, as expected from the very long-tailed degree distributions of the mid-range connectome (*Figure 8—figure supplement 1A*). Nonetheless, this control model still largely underestimates the complexity of the mid-range network. At the same time, the DBM control was the closest control for local connectivity, showing that much of its structure is indeed determined by spatial distance under certain morphological constraints.

## Simplicial cores define central subnetworks, tied together by mid-range connections

Next, we investigated in which layers the neurons forming these high-dimensional structures resided by measuring *node participation*, that is, the number of simplices to which a node belongs, and a measure of the node's centrality (*Sizemore et al., 2018*; see Materials and methods for details). We found that in local connectivity, most simplices of dimensions 2, 3, and 4 have their source in layer 3 and their sink in layer 5 *Figure 8B1* top. Yet, for dimensions 5 and above, we found a shift towards layer 6, with both sources and sinks found mostly in that layer. In mid-range connectivity, the structures

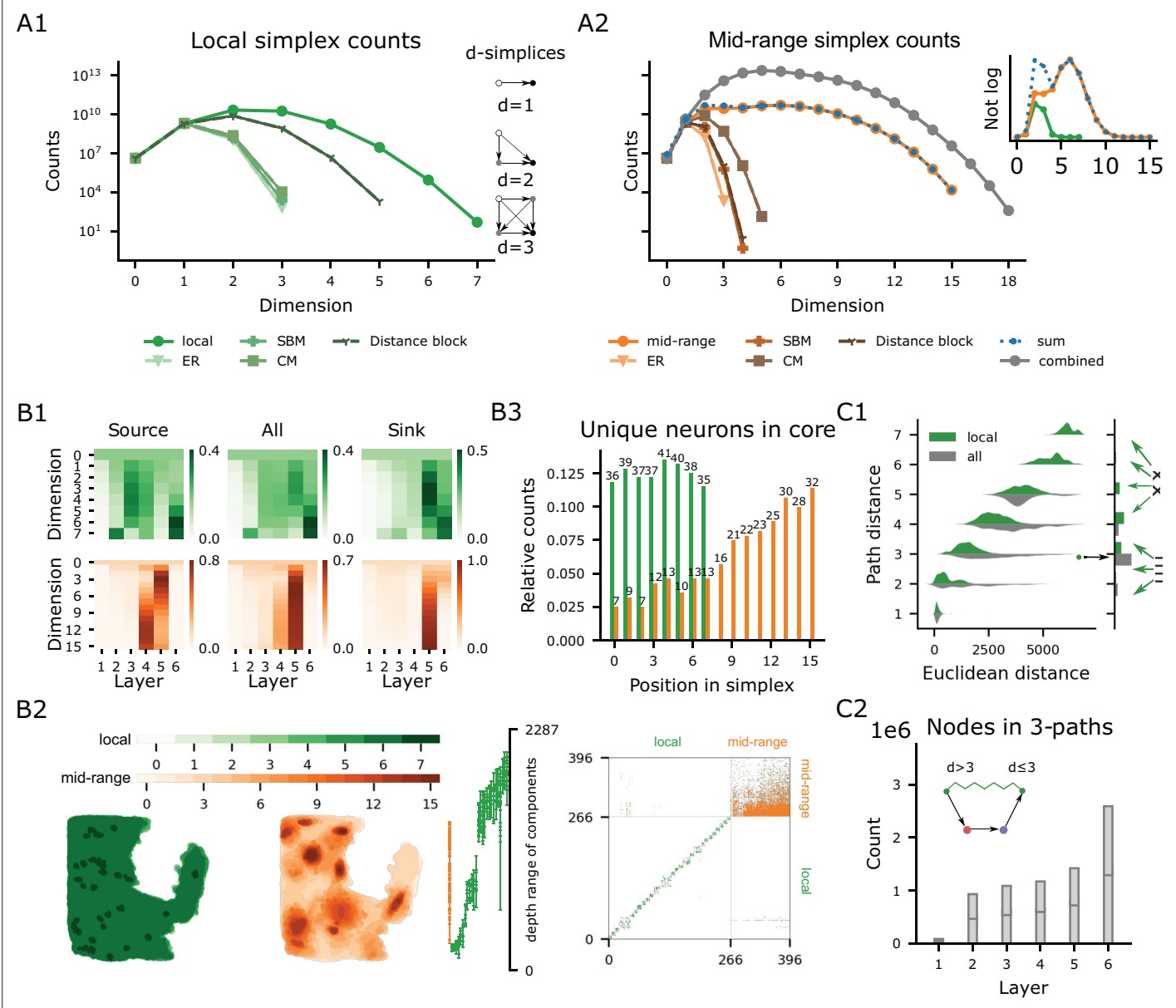

**Figure 8.** Global connectivity structure local vs mid-range. (**A1**) Simplex counts of the local connectivity network and several types of random controls (see text). Examples of $d$-simplices for $d = 1$, 2, and 3. Intuitively, an $n$-simplex is formed by taking an $n - 1$-simplex and adding a new node to which all neurons connect (sink). (**A2**) In orange shades, simplex counts of the mid-range connectivity network and several types of random controls. In gray, simplex counts of the combined network and in blue the sum of the local and the mid-range simplex counts. Inset: local, mid-range and the sum of simplex counts on a linear scale. (**B1**) Node participation per layer, normalized to add up to one in each dimension. Top/bottom row: Local/mid-range connectivity network. Left to right: Participation as source, in any position, and as a sink of a simplex. (**B2**) From left to right: Spatial location of the cells in the simplicial $n$-cores in flat coordinates for all $n$. Depth coordinates for each connected component in the simplicial cores (cells participating in simplices of maximal dimension) each dot marks the depth of a neuron in that component. Adjacency matrix of the simplicial cores with respect to local (green dots) and mid-range (orange dots) connections. (**B3**) Number of unique neurons in the simplicial cores present in each position of a simplex (i.e., position along the order from source to target. numbers: total counts, bars: counts relative to the total). (**C1**) Right: Distribution of path distances between pairs of neurons in the local simplicial core; green: along only local edges; grey: along all edges. Left: Euclidean distances between pairs at a given path distance. Arrows: See C2. (**C2**) Total Number of neurons in all paths of length 3 between neurons of the local core which are at distance 3 in the combined circuit (black arrow in C1) split by location. Red/purple: at positions 2 and 3 respectively; grey: expected from randomly assigned m-types while maintaining the global distribution. Cross-/stripe-patterned: For pairs at path distances greater/less-or-equal than 3 along local edges only (green arrows in C1).

The online version of this article includes the following figure supplement(s) for figure 8:

*Figure 8 continued on next page*

*Figure 8 continued*

**Figure supplement 1.** Basic properties of network connectivity.

**Figure supplement 2.** Rich club analysis.

are much more concentrated on layer 5, which contains both source and sink *Figure 8B1* bottom. Only for simplices of dimension greater than 8 does layer 4 provide more sources. Taken together, this indicates a robust local flow of information from superficial to deeper layers and within deep layers, with layer 5 forming a backbone of structurally strong mid-range connectivity. Neurons in layer 6 form numerous simplices among themselves with no apparent output outside of layer 6, but they are known to be the source of many cortico-thalamic connections (*Shepherd and Yamawaki, 2021*) that are not a part of this model.

The model made an explicit distinction between m-types forming outgoing mid-range connections (i.e. projecting m-types) and those that do not (*Figure 8—figure supplement 1C*, 'proper sinks') leading to bimodal distributions of out-and total degrees in the corresponding DBM and SBM controls (*Figure 8—figure supplement 1A*). In contrast, the actual mid-range network has a unimodal, long-tailed degree distribution, similar to biological neuronal networks (*Giacopelli et al., 2021*). This further demonstrates that the m-types alone can not capture the specific targeting between neuron groups without their actual morphologies.

One type of connectivity specificity that has been found in biological neuronal networks (*Towlson et al., 2013*) is the formation of a rich club (*Zhou and Mondragon, 2004*; *van den Heuvel and Sporns, 2011*). This is characterized by a rich-club curve, which measures whether high degree nodes are more likely to connect together, compared to the CM control (see Materials and methods). We observe that the local network has a rich-club effect, but it is not stronger than expected in a spatially finite model with distance-dependent connectivity (*Figure 8—figure supplement 2B1/2*). On the other hand, the mid-range network does not exhibit a rich-club effect, see *Figure 8—figure supplement 2B2* bottom, showing that high degree nodes are not central to the network.

When we generalized the rich club analysis to higher dimensions by considering node participation instead of degree (*Opsahl et al., 2008*; see Materials and methods for details) a different picture emerged (see *Figure 8—figure supplement 2B3*). We found that the rich club coefficients in the mid-range network increased with dimension, while the opposite happens for the local network. We speculate that this effect persists after normalization with respect to relevant controls. Unfortunately, this cannot be currently verified since generating appropriate random controls for these curves is an open problem currently investigated in the field of random topology (*Unger and Krebs, 2024*). Nonetheless, this indicates that central nodes strongly connect to each other forming a structural backbone of the network that is not determined by degree alone.

We studied these structural backbones, by focusing on the *simplicial cores* of the networks, which is a higher dimensional generalization of the notion of network core, which is determined by degree alone (see Materials and methods). The general trend observed is that the local network is distributed, while the mid-range network is highly localized. First in terms of the locations of neurons participating in the cores (*Figure 8B2* left; green vs. orange), and their laminar locations (*Figure 8B2* right). Second, even though the numbers of neurons in the local and mid-range cores are in the same order of magnitude, the local core has 26 disconnected component while the mid-range core is fully connected (*Figure 8B3* bottom). Finally, the mid-range core was successively more strongly localized towards the source position, where only seven unique neurons were the sources of all of the 15,108 highest-dimensional mid-range simplices (*Figure 8B3* top).

Finally, we studied the interactions between the local and mid-range networks, specifically, the ability of mid-range connnectivity to form short-cuts between neurons of the local core. Even though the sub-network on the nodes in the local core has multiple connected components (*Figure 8B2*), when paths through nodes outside the core are allowed, it is fully connected. When only local connectivity was considered, the path distances between core neurons were widely distributed between one and seven with a median of four and a strong dependence on their Euclidean distance (*Figure 8C1*, green). On the other hand, when mid-range connections were added the maximum path distance drops to five, with a negligible number of pairs at that path distance. The median path distance drops to three, and the dependence of the path distance on the Euclidean distance of the pairs nearly disappears (*Figure 8C1*, grey). Crucially, even though the number of edges almost doubled with the

addition of mid-range connections, the number of direct connections remained negligible; instead paths of length three seem to be dominant for information exchange between neurons.

We therefore studied these paths of length three in the combined circuit in more detail, labeling the neurons along them from position 1 (start-node) to position 4 (end-node). We found that layer 5 (the layer with the highest outgoing edge probability, see *Figure 8—figure supplement 1B*) is over-represented in nodes in positions 2 and 3 compared to a random assignment of m-types (*Figure 8C2*). Moreover, the over-representation depends on the path distance between the start-node and end-node within the local network: when the value is larger than three (bars hatched by crosses), the effect is stronger than for pairs at a distance three or less (bars hatched by horizontal bars). This demonstrates and quantifies how neurons in layer 5 act as 'highway hubs' providing shortcuts between neurons that are far away from each other in the local circuit.

## Discussion

We have presented a model of the non-barrel primary somatosensory cortex of the juvenile rat that represents its neuronal and - particularly - synaptic anatomy in high detail. The model comprises a spatial scale that allows for the study of cortical circuits not only as isolated functional units, but also their interactions along inter-regional connections. It also demonstrates how novel insights that are not readily apparent in disparate data and individual models can be gained when they are combined in a way that creates a coherent whole. Specifically, we were able to make multiple predictions about the structure of cortical connectivity that required integration of all anatomical aspects potentially affecting connectivity. At the scale of this model, anatomical aspects affecting connectivity went beyond individual neuronal morphologies and their placement, and included intrinsic cortical curvature and other anatomical variability. This was taken into account during the modeling of the anatomical composition, for example by using three-dimensional, layer-specific neuron density profiles that match biological measurements, and by ensuring the biologically correct orientation of model neurons with respect to the orientation towards the cortical surface. As local connectivity was derived from axo-dendritic appositions in the anatomical model, it was strongly affected by these aspects.

One type of literature resource our approach to connectivity did not use is experimental measurements of local connection probabilities. While these resources are valuable data for our understanding of cortical processing, they are difficult to use in a biophysically detailed model covering large spatial scales. First, connection probabilities are often identified through a PSP response at the soma (*Song et al., 2005*; *Perin et al., 2011*; *Thomson et al., 2002*). However, for connections formed by low numbers of synapses on distal dendrites the PSP may be attenuated too much to be measurable at the soma (*Neher, 1992*). The connection may still be relevant, e.g., by collaboratively triggering non-linear dendritic events or affecting plasticity of nearby synapses (*Farinella et al., 2014*; *Iacaruso et al., 2017*; *Tazerart et al., 2020*). These effects are not considered in simplified models, leading to a focus on somatically visible connections only. However, in our case, we want to be able to study these effects with our model. Second, some studies report connection probabilities, but only incidentally, being primarily interested in the physiology of a synaptic pathway (*Thomson et al., 2002*; *Markram et al., 1997*; *Mason et al., 1991*). Hence, they employ sampling strategies that aim to maximize the number of connected pairs encountered. Third, even studies aiming to characterize the anatomy of circuitry can provide contradictory estimates: *Jiang et al., 2015* report 0 connections for 150 sampled pairs of PCs in layer 5 of adult mouse V1, while in the proofread portion of the data of *Bae et al., 2021* 13 out of 154 pairs (8.4%) within 100 µm are connected. Fourth, connection probabilities are often measured by sampling pairs of neurons at inter-soma distances below 100 µm or even only 50 µm (see Table 4.1 of *Zhang et al., 2019* for an overview). While connection probabilities drop with distance, in a model covering large spatial dimensions, the number of potentially connected pairs grows with the square of distance. Consequently, a connection probability of, for example 0.15% at 500 µm would represent as many connections as a probability of 15% at 50 µm, and it cannot be rounded to zero. Recently, unbiased connection sampling techniques have been developed that provide estimates at larger distances (*Chou et al., 2023*), that are likely to solve some of these issues in the future.

Connectivity derived from axo-dendritic appositions alone was insufficient at the large spatial scale of the model, as it was limited to connections at distances below 1000 µm. While we found that it generated the right amount of connectivity within a somatosensory subregion, we combined it with a second algorithm for inter-regional connectivity. The algorithm is parameterized by a combination

of overall pathway strength, topographical mapping and layer profiles, which together describe a probability distribution for the segments of mid-range axons, that is, an average axonal morphology, using established concepts. On the dendritic side, the algorithm takes individual neuronal morphologies and their placement into account. We have demonstrated that the sum of local and non-local synapse densities matches the reference. It is important to consider whether such a mixture of two independent algorithms captures potentially important statistical interactions between the local and mid-range connectivity of individual neurons. We note that local connectivity is fully constrained by morphology, and the non-local connectivity is parameterized independently for individual morphological types. The combined connectome therefore captures important correlations at that level, such as stronger and weaker non-local cortico-cortical connections from slender-tufted and thick-tufted layer 5 PCs, respectively. When the model was created, interactions beyond this level had not yet been described because local axon reconstructions are reconstructed from slices, which prevents the possibility of obtaining long range information. Even in vivo staining still only reconstructs axons within a region (*Buzás et al., 2006*). On the other hand, long-range axon reconstructions are obtained through whole brain staining, which has low accuracy for local connectivity (*Winnubst et al., 2019*). Analysis of new EM datasets, such as a characterization of distance-dependent targeting of excitatory/inhibitory neurons by the axons of L5 thick-tufted pyramidal cells (*Bodor et al., 2023*) using the data of *Bae et al., 2021* could be incorporated in future models. Finding non-random correlations between local and non-local connections would require a strong null model to compare measurements to and our model can serve as that.

Recreating electron-microscopic analyses of connectivity in silico, we could then predict limitations of predicting connectivity from neuron placement and morphology and characterize the additional mechanisms shaping connectivity. Conceptually, a number of mechanisms determine the structure of synaptic connectivity, which we will list from general and large-scale to specific and micro-scale: First, large-scale anatomical trends over hundreds of µm, given by non-homogeneous (e.g. layered) soma placement and broad morphological trends (e.g. ascending axons). These are trends that can be captured by simple distance or offset-dependent connectivity models (*Gal et al., 2020*). Second, small-scale morphological trends captured by axonal and dendritic morphometrics such as branching angles and tortuosity. These are trends that require the consideration of individual morphologies and their variability instead of average ones. Third, the principle of cooperative synapse formation (*Fares and Stepanyants, 2009*) formalizing an avoidance of structurally weak connections. Fourth, any type-specific trends not captured by neuronal morphologies, such as local molecular mechanisms and type-specific synaptic pruning. Fifth, non-type specific synaptic rewiring, for example through structural plasticity. We will refer to these aspects as (L)arge-scale, (S)mall-scale, (C)ooperativity, (T)ype-specificity and (P)lasticity respectively, and we argue that for the explanation of the connectome, the more general explanations should be exhausted first, before moving on to more specific ones. Also note that this classification is not considering the underlying developmental *causes*, but instead associates anatomical and non-anatomical predictors with the structure of the connectome. (L) has been shown to accurately predict large-scale connectivity trends, giving rise to *Peters' rule* (*Peters and Feldman, 1976*; *Garey, 1999*, although the exact meaning of Peters' rule is debated, see *Rees et al., 2017*). It has been demonstrated that (L) alone does not suffice to explain non-random higher order trends in excitatory connectivity, while the combination of (L,S,C) does (*Gal et al., 2017*; *Reimann et al., 2017a*; *Gal et al., 2020*; *Udvary et al., 2022*). More specifically, *Reimann et al., 2017b* found that (L,C) explains overexpression of reciprocal connectivity in cortical circuits, but (L,S,C) is required to match the biological trend for clustered connectivity, and *Udvary et al., 2022* found (L,S) recreates non-random triplet motifs. *Billeh et al., 2020* combined (L) with highly refined (P)-type rules, demonstrating great *functional* match of the resulting model.

Our comparison to the results of *Motta et al., 2019* cannot capture the role of (L), as the scale of the considered volume is too small. But it demonstrates that some non-random targeting trends can be explained by (S,C) and highlights the importance of (C). Further, the overall lower specificity of excitatory axon fragments indicates that (T) may not play a role for them. Similarly, the comparison to *Schneider-Mizell et al., 2024* shows that (T) is not required to explain the connectivity of 'distal-targeting' (i.e., Sst+) neuron types. Conversely, for 'perisomatic-targeting' types (i.e., PV+ basket cells) (T) was required to match the distribution of postsynaptic compartment types. Additionally, the number of potential synapses remaining after applying an optimized (T)-type mechanism was

**Table 2.** Anatomical, morphological and other aspects affecting connectivity and our predictions for their relevance for efferents of different neuron types.

See main text for an explanation of the individual aspects. The signs in the table indicate whether we predict a certain aspect to be not relevant (-), relevant (+), or highly relevant (++) for a given neuron type.

| Axon type | (L)arge-scale morphological | (S)mall-scale morphological | (C)ooperativity | (T)ype-specificity | (P)lasticity |
|---|---|---|---|---|---|
| EXC. | + | ++ | ++ | - | + |
| BCs | + | + | + | + | + |
| Sst+ | ++ | + | + | - | + |
| VIP+ | + | - | + | ++ | - |
| NGC and L1 | ++ | + | - | +/- | + |

too large to be sustained by the axons, implying a crucial role of (P) in reducing it further. This is in contrast to 'inhibitory-targeting' neurons, where an optimized (T)-type pruning lowered the synapse count so much that no space for (P) remained. For the 'sparsely-targeting' neurons of *Schneider-Mizell et al., 2024* we developed two competing hypotheses, one predicting no role for (T) and 70% of their synapses volumetrically transmitting, that is, without clear postsynaptic partner, and the other predicting a role for (T) similar to the perisomatic-targeting neurons. We summarize our predictions in *Table 2*.

In addition to targeting specificity, we predict the following: First, we were able to predict the effect of cortical anatomical variability on neuronal composition by increasing or decreasing the space available for individual layers. If we assume that each layer has a given computational purpose (*Felleman and Van Essen, 1991*), then this may have functional consequences. It is possible that cortical circuits compensate for this effect, either anatomically (e.g. using different axon or dendrite morphologies), or non-anatomically. Either case would imply the existence of an active mechanism with the possibility of malfunction. Alternatively, function of cortical circuits is robust against the differences in wiring we characterized. This can be studied either in vivo, or in silico based on this model.

Second, we predicted constraints on the emergence of cortical maps from the anatomy of thalamo-cortical innervation, that is, from the combination of: shape and placement of cortical dendrites, the specific layer pattern formed by thalamo-cortical axons, and their horizontal reach. We demonstrated how differences in these parameters may affect the distances between neuron pairs with similar stimulus preferences. At the lower end, very different preferences are supported even between neighboring neurons. At the higher end, neurons further than ~350 μm apart are likely to sample from non-overlapping sets of thalamic inputs. Other mechanisms will ultimately affect this - chief among them structural synaptic plasticity - this can be thought of as the ground state that plasticity is operating on.

Third, we predicted the topology of synaptic connectivity with neuronal resolution at an unprecedented scale, that is, combining local and mid-range connectivity. We found that the structure of neither can be explained by connection probabilities, degree-distributions, or distance-dependence alone, not even when individual pathways formed between morphological types are taken into account. We expect the mid-range network to have a small-world topology, but computing small-world coefficients for a network of this size is infeasible (see Materials and methods).

Fourth, we predict that the mid-range connectivity forms a strong structural backbone distributing information between a small number of highly connected clusters. The paths within the mid-range network strongly rely on neurons in layer 5 and form short-cut paths for neurons further away than ~2 mm; for smaller distances, local connectivity provides equivalent or shorter paths.

All these insights required the construction of an anatomically detailed model in a three-dimensional brain atlas; additionally, most of them required the large spatial scale we used. Their functional and computational implications are difficult to predict without also considering data on neuron activity. To that end, an accompanying manuscript describes our modeling of the physiology of neurons and synapses, along with a number of in silico simulation campaigns and their results. Additionally, more insights were gained from using the model in several publications: In *Ecker et al., 2024b*; *Ecker*

*et al., 2024a*; *Egas Santander et al., 2024a* it has been demonstrated that non-random higher order structure affects functional plasticity, assembly formation and the reliability and efficiency of coding of subpopulations. Moreover, two of these insights were validated against electron microscopy data. Furthermore, in *Reimann et al., 2024* the model served as an important null model to compare an electron microscopic connectome to, enabling the discovery of specific targeting of inhibition at the cellular level.

This iteration of the model is incomplete in several ways. First, several compromises and generalizations were made to be able to parameterize the process with biological data, most importantly, generalizing from mouse to rat. We note that such mixing is the accepted state-of-the-art in the field, and also neuroscience in general. All 19 data-driven models of rodent microcircuitry listed in *Figure 2* of the recent review of *Ramaswamy, 2024* conduct some sort of mixing, including the very advanced mouse V1 model of the Allen Institute (*Billeh et al., 2020*). In this iteration we focused on investigating the general features of the (multi-region) mammalian cortex, e.g., high-order motifs, connected by L5 neurons across subregions or the effect of curvature on connectivity. In the future, more specific aspects of different cortical regions could be investigated using more specific data sources. This would lead to different versions of the model for different regions that can be compared and contrasted using the various structural metrics we developed in this work. Further, we made a number of assumptions about the biological systems that we explicitly list in *Supplementary file 2*. Should improved and more specific data sources, such as EM-based connectomes or whole-brain neuron reconstructions, become available in the future, our modeling pipeline is designed to readily utilize them for refinement. There are also known limitations to the actual model building steps. While we enforce cell density profiles along the depth-dimension, densities are assumed to be otherwise uniform within a given subregion. For additional structure, the algorithms will need to be updated; this will be required for example, for the modeling of the barrel field. Similarly, during volume filling, orientation and placement of neuron morphologies is considered only along the depth-axis. Modeling of the barrel system will also require the consideration of orientation and location with respect to nearby barrel centers. Furthermore, we assumed uniform relative layer thickness throughout the modeled region, due to sparsity of the required data. However, given that *Yusufoğulları et al., 2015* has demonstrated there are substantial differences in layer thickness between rat hindlimb and barrel field regions and *Narayanan et al., 2017* and *Wagstyl et al., 2020* found lower but still significant differences within developed rat barrel cortex and human somatosensory cortex respectively, the model pipeline should be updated once sufficient data becomes available.

This demonstrates that detailed modeling requires constant iteration, as a model can never be proven to be right, only to be wrong. Thus, we made the tools to improve our model also openly available (see Data and Code availability section). *Figure 1* provides a technical overview of individual modeling steps, the software tools involved and the duration required to run them, to serve as a guide to interested contributors.

## Materials and methods

**Key resources table**

| Reagent type (species) or resource | Designation | Source or reference | Identifiers | Additional information |
|---|---|---|---|---|
| Software, algorithm | Model loading and interaction | https://doi.org/10.5281/zenodo.8026852 | | |
| Software, algorithm | Model analysis | https://doi.org/110.5281/zenodo.8016989 | | |
| Software, algorithm | Topological analysis of connectivity | *Egas Santander et al., 2024b*; https://github.com/danielaegassan/connectome_analysis | | see also *Egas Santander et al., 2024a*; version 0.0.1 |
| Software, algorithm | atlas-splitter | *Gevaert et al., 2024a*; https://github.com/BlueBrain/atlas-splitter/tree/v0.1.5 | | Release 0.1.5 |
| Software, algorithm | atlas-densities | *Gevaert et al., 2024b*; https://github.com/BlueBrain/atlas-densities/tree/v0.2.5 | | Release 0.2.5 |
| Software, algorithm | Brainbuilder | *Povolotsky et al., 2024*; https://github.com/BlueBrain/brainbuilder/tree/v0.20.0 | | Release 0.20. |

*Continued on next page*

4000

*Continued*

| Reagent type (species) or resource | Designation | Source or reference | Identifiers | Additional information |
|---|---|---|---|---|
| Software, algorithm | NeuroR | *Arnaudon et al., 2024*; https://github.com/BlueBrain/NeuroR/tree/v1.7.0 | | Release 1.7.0 |
| Software, algorithm | NeuroC | *Gevaert et al., 2024c*; https://github.com/BlueBrain/neuroc/tree/v0.3.0 | | Release 0.3.0 |
| Software, algorithm | placement-algorithm | *Povolotsky et al., 2023*; https://github.com/BlueBrain/placement-algorithm/releases/tag/placement-algorithm-v2.4.0 | | Release 2.4.0 |
| Software, algorithm | Appositionizer | *Wolf and Berchet, 2024*; https://github.com/BlueBrain/appositionizer/releases/tag/v1.0.0 | | see also *Kozloski et al., 2008*; release 1.0.0 |
| Software, algorithm | Functionalizer | *Wolf et al., 2024*; https://github.com/BlueBrain/functionalizer/releases/tag/v1.0.0 | | Release 1.0.0 |
| Software, algorithm | Projectionizer | *Gevaert and Herttuainen, 2022*; https://github.com/BlueBrain/projectionizer/releases/tag/projectionizer-v3.0.0.dev0 | | Release 3.0.0.dev0 |
| Software, algorithm | Connectome-Manipulator | *Pokorny et al., 2024b*; https://github.com/BlueBrain/connectome-manipulator/tree/v1.0.0 | | see also *Pokorny et al., 2024a*; release 1.0.0 |

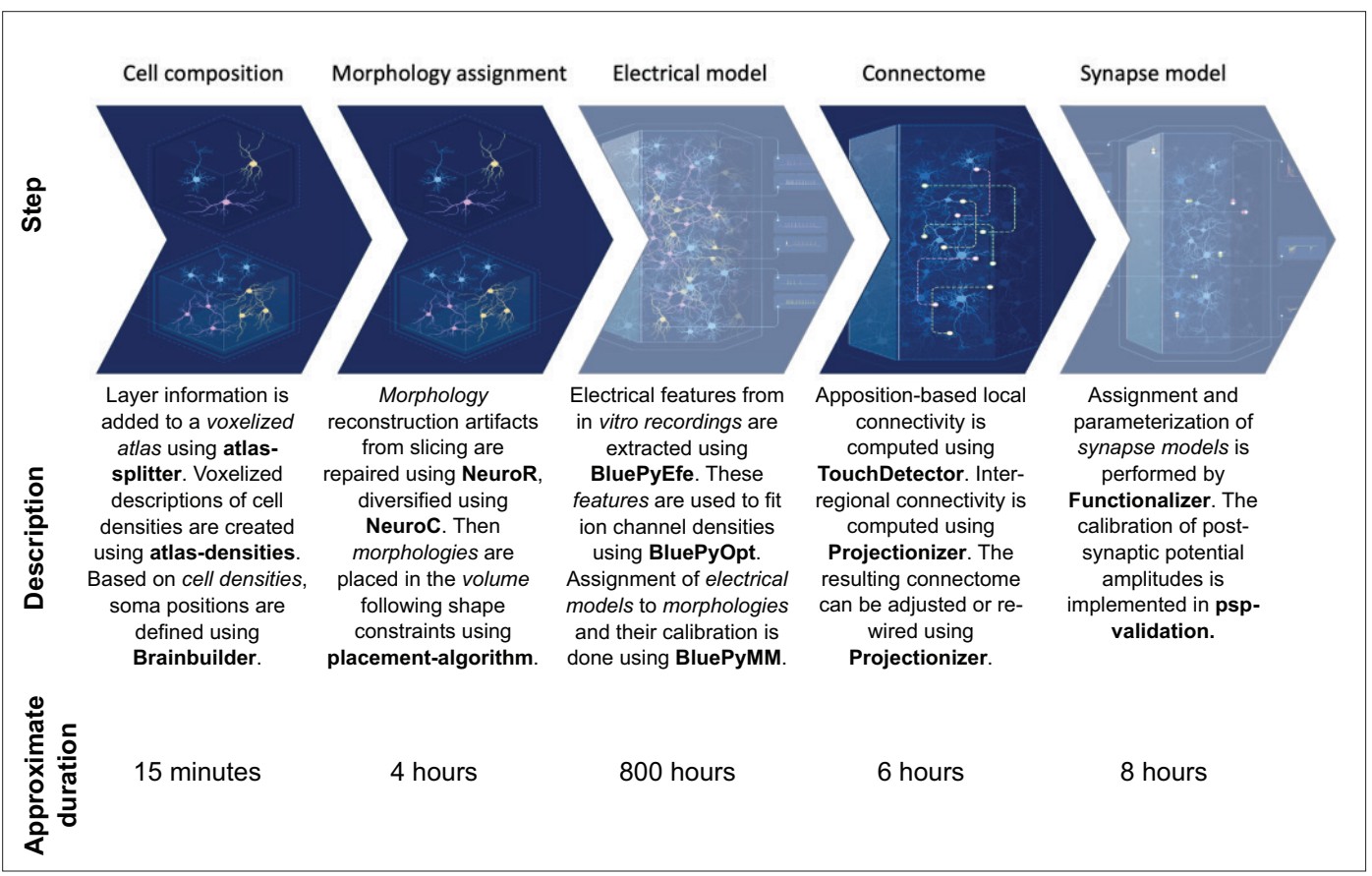

**Figure 9.** Overview of the building workflow and tools used. Names of individual software tools are bold in the description (see also Key resources table); descriptions of required inputs in italics. Where not listed, the inputs of a step are outputs of the previous step. For completeness, also physiological modeling steps not described in this work are also depicted, but indicated semi-transparent. Note that the Key resources table only lists tools related to anatomical modeling.

## Method details

An overview of the model building steps, tools used and approximate durations can be found in *Figure 9*.

### Preparation of voxelized atlases

First and foremost, the departure from a simplified hexagonal geometry (as in the NMC model *Markram et al., 2015*), required a digital brain atlas defining the anatomy of the region. To that end, we took as starting point the *Paxinos and Watson, 2007* adult rat brain atlas, processed individual digitized slices, aligned them and labeled them to assemble a smooth three-dimensional volume with region annotations. The resulting atlas was then scaled down from the dimensions of adult rat to juvenile (P14) rat brain by reducing the size of individual voxels from 40 µm to 38.7348 µm. The scaling factor was based on the ratio of S1HL thicknesses at those ages (2082 µm for juvenile vs. 2150 µm for adult). Finally, supplementary atlas datasets were generated as described in *Bolaños-Puchet et al., 2024*; *Bolaños-Puchet et al., 2023*. These datasets provide additional spatial information required to ensure biologically accurate placement and orientation of dendritic trees (see below). First, the normalized cortical depth (between 0 and 1) at each point; second, the local orientation as a vector pointing towards the cortical surface at each point; third, the total cortical thickness at each point, that is, the length of the shortest path from the cortical surface to the bottom of layer 6 passing through that point.

Additionally, we followed *Bolaños-Puchet et al., 2024*; *Bolaños-Puchet et al., 2023* to produce a *flat map* of somatosensory regions, that is, a coordinate transformation associating each voxel with a two-dimensional projection into the plane. This can be used to create a flat view of region annotations that is crucial for the description of the topographical mapping of mid-range synaptic connectivity (see below). In short, to produce the flat map, a projection surface was defined by reconstructing a mesh from all points at a relative cortical depth of 0.5. Next, the local orientation field in the supplementary atlas was numerically integrated, yielding streamlines that were used to project each voxel center onto the projection surface. Finally, the projection surface was flattened with an authalic (area-preserving) algorithm. The main property of the resulting flat map is that in any flat view derived from it, each pixel represents a subvolume of cortex that spans all layers, akin to a cortical column.

### Reconstruction of neuron morphologies

Neuronal reconstructions were collected with two techniques: neurons were either filled with biocytin in a brain slice and reconstructed (*Markram et al., 2015*; *Muralidhar et al., 2013*), or were filled with a fluorescent dye in vivo and reconstructed (*Buzás et al., 1998*; *Karube and Kisvárday, 2011*). In total 1017 unique reconstructions were used, 896 of which were previously used in *Markram et al., 2015*, 63 were new in vitro reconstructions, and 58 were new in vivo stained reconstructions.

### Animal surgery

Normal C57BL/6 mice (P52-60) were used which were bred and maintained in the animal house facility of Department of Anatomy, Physiology and Embryology under appropriately controlled conditions (approval of Local Ethics Committee for Animal Research Studies at the University of Debrecen in line with European Union guidelines for the care of laboratory animals, Directive 2010/63/EU).

For initial anesthesia, animals were injected with pentobarbital (0.15 ml (6 mg/ml), i.p.). Prolonged anesthesia was achieved by injecting 0.05 ml every 60–90 minutes depending on the reaction to the toe-test. Head restraining was used and craniotomy performed in both hemispheres at coordinates ML 1.5–2 mm; AP 0.5–1 mm (*Paxinos and Watson, 2007*) in order to expose the primary somatosensory cortex at the representation of the (hind-limb and fore-limb). We applied the topical anesthetic Lidocaine (Xylocaine gel, Egis Gyógyszergyár ZRT, Hungary) to all surgical wounds and pressure points. A custom-made plastic chamber (prepared by cutting a ring from a 1 ml plastic syringe) was mounted on the skull using super glue and was surrounded by 4% agar (VWR International Kft, Hungary). Then, the cortex was exposed by making a slit on the dura mater with the bent tip of a 14 gauge hypodermic needle.

## Single neuron labelling

For this purpose, borosilicate glass pipettes (GB150F-8P, Science Products GmbH, Germany) were pulled (Model P-97, Sutter Instrument Co., USA) with a resistance in the range of 60–80 MΩ (bevelled on a BV-10M beveler, Sutter Instrument Co., USA) and filled with 1 M KCl containing 2% biocytin (Sigma-Aldrich Chemie GmbH, Germany). The microelectrode was attached to a hydraulic micro-drive (MHW-4 Narishige, Japan) and the tip guided in the cortex under the guidance of a surgical microscope (OP-MI, Zeiss). Then the chamber was then filled with 4% agar (VWR International Kft, Hungary) for better stability of the micropipette. Neuronal activity was recorded and amplified with AxoClamp-2A (Axon Instruments Inc, USA). After filtering, the signal was displayed on an oscilloscope and an audio monitor to aid and control intracellular penetration. Successful penetration of the cell membrane was indicated by a sudden drop of the resting membrane potential (below –40 mV) while applying 0.05 nA in the step-current mode (put here stimulus configuration from Master-8). Then, biocytin was injected with +2 nA using a duty cycle of 400 msec on and 200 msec off typically for 20 min. In each hemisphere 1–3 penetrations were made with approximately 0.3 mm spacing from each other. Neuronal activity was searched blindly across the entire cortical depth.

## Histology

The animals received a lethal dose of anesthetics and were perfused transcardially first with the washing medium (oxygenated Tyrode's solution) for 2 min or until the blood showed clearing and then with a fixative (approx. 100 ml) containing 2% paraformaldehyde (VWR International Kft, Hungary) and 1% glutaraldehyde (Sigma-Aldrich Chemie Gmbh, Germany) in 0.1 M phosphate buffer (PB, pH 7.4) for 50 minutes. Next, the brain was removed from the skull and tissue blocks containing the region of interest were dissected. A series of 60–80 um thick vibratome (Leica VT1000S, Leica Biosystem) sections were cut in the coronal plane and collected in 5x10 lots in glass vials. The sections were washed in 0.05 M Tris-buffer saline (TBS, 10 min) and 0.05 M TBS containing 0.1% Triton X-100 (2x10 min) and incubated at 4 °C in avidin-biotin complexed-HRP (ABC-Elite kit, Vector Laboratories, Inc, USA), diluted 1:200 in 0.05 M TBS containing 0.1% Triton X-100 (Sigma-Aldrich Chemie Gmbh, Germany) for overnight. Then the sections were washed in TBS for 2x10 min and in TB for 10 min. They were treated with 0.05% DAB (3.3' diaminobenzidine-4HCl, Sigma-Aldrich Chemie Gmbh, Germany) diluted in 0.05 M TB containing 0.0025% CoCl2 for 30 min while agitating on an electric shaker. Finally the labelling was visualized in the presence of 0.1% H2O2 (5 min to 10 mins). After washing the sections in 0.05 M TBS (3x10 min) and 10 min in 0.1 M PB the quality of the DAB reaction and the presence of intracellularly labelled cells were inspected while wet under a light microscope (x10 objective). All sections of blocks containing strongly labelled neurons underwent osmification, dehydration and resin embedding in order to retain the 3D structure of their axons and dendrites. Accordingly, sections were treated with 1% OsO4 (osmium tetroxide, PI Chem Supplies, USA) diluted in 0.1% PB for 15 minutes. After rinsing in 0.1 M PB for 3x10 min, they were dehydrated in ascending series of ethanol (50, 70, 90, 95 per cent and abs ethanol), propylene oxide, each step for 2x10 min, and submerged in resin (Durcupan, Sigma-Aldrich Chemie GmbH, Germany) overnight at room temperature. Finally, the sections were mounted on glass slides, coversliped and cured at 56 oC for 24 hours (*Somogyi and Freund, 1989*).

## Morphological neuron reconstruction

Labelled neurons were reconstructed in 3D using the Neurolucida neuron reconstruction system running on Windows XP platform (Neurolucida v.8.23, MicroBrightField Inc, Williston, USA). For this purpose a Leica DMRB microscope (x100 objective) was coupled to a motorized XY-stage and a z-motor via a stage controller (Märzhäuser Wetzlar GmbH & Co. KG, Wetzlar, Germany). Each neuron was reconstructed from 20 to 32 adjoining sections. Neighboring sections were aligned using the 3-point alignment and the least-squares algorithm for the cut ends of labelled processes and fiducial landmarks such as the contour of cut blood vessels. The cell body, dendrites, axons and axon terminals were reconstructed together with their thickness value.

## Morphology curation and classification

All morphological neuron reconstructions (see Reconstruction of neuron morphologies) were curated and repaired to correct reconstruction errors and slicing artifacts, as described in *Markram et al.,*

*2015*. They were then classified based on the following strategy. Both interneurons and pyramidal cells were first classified by expert reconstructors according to their observed shapes by inspection through the microscope. The expert classification of pyramidal cells, which is based on the shape of the apical dendrites, was then used as input for the training of the algorithms for the objective classification as presented in *Kanari et al., 2019*. The objective classification was performed based on the *topological morphology descriptor* (TMD) (*Kanari et al., 2018*), which encodes the branching structure of neuronal trees. The TMD of apical trees was extracted from all morphological reconstructions and was used to train different classifiers for the objective classification of cells into distinct groups. See *Figure 2—figure supplement 3* for exemplary excitatory morphologies and their features.

The expert-proposed scheme comprised 60 morphological types (m-types) (18 excitatory and 42 inhibitory). The m-types of pyramidal cells are distinguished first by layer and further by shape of their tuft, such as untufted (UPC) and tufted (TPC) cells; and finally into subclasses (A:large tufted, B:late bifurcating, C:small tufted). The results of the objective classification of pyramidal cells were then used to validate this classification scheme. A classification scheme was deemed valid if the TMD was significantly different between classes. This was the case for classes in all layers except for a distinction between two subgroups in layer 3 (L3_TPC:A vs. L3_TPC:B), which was consequently discarded. Instead we performed an unsupervised clustering of all tufted pyramidal cells in layer 3 based on their TMD and found a different split into two classes, best described as large tufted (L3_TPC:A) and small tufted (L3_TPC:C), which we then used in the model. The resulting list of m-types is found in .

For the remaining layers, the results of the objective classification were published in a paper *Kanari et al., 2024* describing a variety of methodologies, using TMD *Kanari et al., 2018* in combination with advanced machine learning tools, such as *Convolutional Neural Networks* and *Graph Neural Networks*. The expert classification could be supported by a variety of methods, ranging from 60% in interneurons to 80-90% in pyramidal cells. Consequently, we considered the expert classification to be sufficiently accurate to build the model. To increase the morphological variability in the model we combined axon reconstructions with soma and dendrite reconstructions from other neurons of the same m-type (*mix & match*; *Markram et al., 2015*).

## Preparation of cell density data

Inhibitory cell densities were constrained following *Keller et al., 2019*. In brief, several datasets were combined to provide depth profiles of densities for successively more granular neuron classes (*Figure 1—figure supplement 1*). The first dataset consists of neuronal soma density estimations, using antibody stains of neuronal nuclear protein (NeuN) and $\gamma$-aminobutyric acid (GABA) from rat neocortex (n=6, *Keller et al., 2019*). Cell counts provided mean densities and a measure of inter-individual variability (*Markram et al., 2015*). Cell counts were obtained from a single rat in this dataset, and their positions were annotated and divided into 100 equal-width bins extending from the top of layer 1 (L1) to the bottom of layer 6 (L6) (*Keller et al., 2019*). This provided depth profiles of both total neuron (from NeuN) and inhibitory neuron densities (from GABA). A similar profile was also estimated from immunostaining with calbindin (CB), calretinin (CR), neuropeptide Y (NPY), parvalbumin (PV), somatostatin (SOM) and vasoactive intestinal peptide (VIP) (at least three slices from at least two rats) (*Keller et al., 2019*). All stains were corrected for shrinkage (*Ghobril, 2015*). A single-neuron reverse transcription polymerase chain reaction (RT-PCR) dataset (*Toledo-Rodriguez et al., 2005*) allowed mapping of biochemical markers to morphological types, by finding spatial distributions of morphological cell types that would reproduce the biochemical marker distribution (*Keller et al., 2019*). Whenever the classification could not be resolved down to morphological types (i.e., excitatory cells, neurogliaform cells, L1 cells, etc.), an estimation of the fractions of subtypes was used (*Muralidhar et al., 2013*). The depth profile of excitatory neuron densities was further subdivided into subtypes, maintaining the same layerwise proportions of these types as in previous work (*Markram et al., 2015*). The final output of this process was a dataset of neuron density as a function of cortical depth for each of the 60 morphological types.

## Voxelized neuronal composition

To prepare the modeled volume for cell placement, we first associated each voxel of the atlas with a cortical layer. We assumed layer boundaries to be at the same normalized depth at each point of the region. The depths, derived from *Markram et al., 2015*, are listed in *Supplementary file 5*. As a

voxel atlas of normalized depths was provided as an input, layer identities could be readily looked up from that table. Similarly, neuron densities for each voxel of the atlas were produced from the vertical density profiles that served as inputs (see Preparation of cell density data) by looking up the corresponding values using the normalized depth.

## Neuron placement

Neurons were placed into the volume by first generating soma positions and annotating them with a morphological type according to the voxelized densities generated. Next, for each location we selected a reconstructed morphology from the annotated morphological type. As previously (*Markram et al., 2015*), we tried to select an appropriate reconstruction taking also the variability within a morphological type into account. For instance, the largest exemplars of layer 5 pyramidal cells cannot be placed close to the top of the layer, as otherwise their tuft would stick out of the top of layer 1. We selected morphologies by scoring them according to how well manually identified features, such as dendritic tufts or horizontal axonal branching, would land in the biologically appropriate layers, when placed at any given location. The placement rules that were used were the same as in *Markram et al., 2015* and are listed in *Supplementary file 6*.

Previously, this process was aided by the use of a simplified geometry where layer boundaries were formed by parallel planes. To execute the algorithm in a realistic brain volume, we used auxiliary voxel atlasses (see Preparation of voxelized atlases). The first atlas contained for each voxel the normalized depth of its center point, that is, a value between 0 and 1 where 0 would indicate its placement at the top of layer 1 and 1 the bottom of layer 6. The second atlas contained for each voxel the total thickness of the cortex at that location. As layer boundaries in the model were always placed at fixed normalized depths (*Supplementary file 5*), we could calculate the absolute distance of the voxel to any layer boundary by subtracting their normalized depths and multiplying the result with the local cortical thickness. Based on this, we calculated the overlap of the dendritic and axonal features of a candidate morphology with the target layer interval (*Figure 2—figure supplement 4*). Morphology selection was then performed as previously (*Markram et al., 2015*), that is, a morphology was selected randomly from the top 10% scorers for a given position.

## Modeling synaptic connectivity

### Local connectivity

To determine the structure of synaptic connectivity between neurons, data on numbers of synapses and their locations were required. Previous data for mean bouton densities on axons of various neuron types, and number of synapses per connection (*Reimann et al., 2015*), were used and generalized to all nbS1 subregions. The data was then used as described in *Reimann et al., 2015*. Similar to neuronal morphologies (see above), there is no evidence for anatomical differences in connectivity between nbS1 subregions, with the exception of barrel cortex (not included in the model).

### Mid-range connectivity

Due to the larger spatial scale of the present model compared to *Markram et al., 2015*, additional data was required to further constrain synaptic connectivity at a global scale. To that end, we referred to data on relative strengths of synaptic connections from the Allen Mouse Brain Connectivity Atlas (*Harris et al., 2019*), and generalized it for use with a rat model. We began by scaling the relative connection densities of *Harris et al., 2019* to absolute densities in units of synapses per $\mu m^3$. That is, the entire voxel-to-voxel connection matrix of intra-cortical connectivity was scaled to match the average total density of synapses measured in electron microscopy (*Schüz and Palm, 1989*).

Next, we summed the densities over voxels belonging to pairs of nbS1 subregions, resulting in a 6 x 6 connection matrix of synapse densities in pathways between and within regions. We mapped this matrix to rat nbS1 by finding corresponding regions in the mouse and rat atlases (*Figure 4—figure supplement 1A*). Here, we assumed that synapse densities in these pathways are comparable between mouse and rat, although the larger dimensions of the rat brain resulted in larger absolute synapse counts (*Figure 4—figure supplement 1C*).

We used the same mapping to generalize the spatial structure of the targeting of connections between regions from mouse to rat. This refers to the question of which specific parts of a region are innervated by individual axons in a different region. *Reimann et al., 2019* modeled this innervation as

a topographical mapping between pairs of regions in a flat view of neocortex. We used a flat map (see Preparation of voxelized atlases) to create a flat view of rat nbS1 subregions and recreated a matching topographical mapping between them as follows (see *Figure 4—figure supplement 1A*).

First, three points were identified inside the flat view of a mouse region such that the area of the enclosed triangle is maximized. Color labels ('red', 'green' and 'blue') are arbitrarily assigned to the three points. Next, another three points defining a maximal area triangle were placed in the flat map of the corresponding rat region(s). The same color labels were then manually assigned to these points to best recreate their spatial context in the mouse triangle, e.g., if the point in SSp-n closest to the SSp-bfd border was labeled 'red', then the point in S1ULp closest to the S1BF border will also be labeled 'red'. We then assume that any point in a rat region corresponds to the equally labeled point in the corresponding mouse region, with linear interpolation and extrapolation between points (*Figure 4—figure supplement 1B*). This assumption transplants the existing prediction of the topographical mapping from mouse to rat, resulting in the mapping depicted in *Figure 4—figure supplement 1D*. We scaled the mapping variance by a factor of 2 to account for the larger size of S1 in our rat flatmap (160 flat units) than in the mouse flatmap (80 flat units).

Finally, we derived predictions for the relative distributions of synapse locations across cortical layers for connections between subregions. *Reimann et al., 2019* provided predictions for all pairs of source and target regions in mouse, which we applied to the corresponding pairs of subregions of rat nbS1. That is, we assumed these layer profiles generalize to rat, albeit using the thicker cortical layers of rats (*Figure 4—figure supplement 1E1 vs E2*).

These constraints were then used to build mid-range connections as described in *Reimann et al., 2019*.

## Preparation of thalamic input data

As previously described in *Markram et al., 2015*, we aimed to add synaptic connections from thalamic regions to the circuit model. While they will ultimately serve as one of the controllable inputs for the simulation of in vivo-like experiments, they can also be used to predict innervation strengths in our anatomical model. Specifically, we modeled two such inputs: one 'bottom-up' input with a *core*-type laminar profile, and one 'top-down' input with a *matrix*-type laminar profile (*Harris et al., 2019*; *Guo et al., 2020*; *Shepherd and Yamawaki, 2021*). To create biologically realistic thalamo-cortical projections with a meaningful distinction between core- and matrix-type inputs, we needed data on the strengths of these pathways and the laminar profiles of their synapses.

As in *Markram et al., 2015*, we used data on cortical innervation by thalamic sources from *Meyer et al., 2010*, which yielded information on the depth profiles of thalamo-cortical projections in barrel cortex. To extrapolate to non-barrel somatosensory cortex, we considered data for the VPM-S1BF pathway to be representative of core-type inputs, and generalized it to all nbS1 regions (*Figure 6—figure supplement 1A1*, left). Similarly, we considered data for the POm-S1BF pathway as representative of matrix-type inputs (*Figure 6—figure supplement 1A2*, left), and generalized it to all nbS1 regions. Since the data is reported as absolute volumetric bouton density of thalamo-cortical axons, we were able to derive the total number of synapses to place by assuming one synapse per bouton and summing over the entire innervated volume. The depth profiles for both pathways featured two clearly separated peaks. We digitized the depth profiles and split them into 10 bins per peak. Furthermore, we applied a threshold of $0.01/\mu m^3$ below which values were set to 0 (*Figure 6—figure supplement 1A*).

To constrain the innervation strength and targeting of individual thalamic axons, we used morphological reconstructions of thalamo-cortical projection neurons from the Janelia MouseLight project ( mouselight.janelia.org; *Economo et al., 2016*). This is a generalization of mouse data to a rat model, made necessary by the lack of a comparable resource for rat. To parameterize core-type projections, we calculated the total axonal length in somatosensory areas of $n = 11$ reconstructions with somata in VPM and axons reaching the somatosensory areas. We found lengths between 5 and 60 mm (*Figure 6—figure supplement 1B*) with a median of 27 mm that we combined with an assumed synapse density of 0.2/μm *Reimann et al., 2015* to get an average number of 5400 synapses per projection fiber. To estimate the lateral spread of the area innervated by individual axons (the vertical component is covered by the layer profiles), we considered the locations of reconstructed axon segments contained within the somatosensory regions. We then fit a Gaussian to the lateral distance

of the segments from their center (*Figure 6—figure supplement 1C*), resulting in a median value of 120 µm.

Unfortunately, the MouseLight database contained only a single neuron with its soma in the POm region and its axon reaching somatosensory areas (labeled as AA604). Visual inspection revealed that its axon mostly avoided these areas to target more medial motor-related regions. As such, to parameterize the matrix-type projections, we instead calculated the lateral spread of the single axon in the motor areas (300 µm in MOp; 172 µm in MOs; mean: 236 µm). Its total length in cortical regions was 28 mm.

## Modeling the structure of thalamo-cortical innervation

We determined the dendritic locations of thalamo-cortical input synapses as previously described in *Markram et al., 2015*, but adapted for the more complex geometry of this model. Briefly, binned depth profiles of densities of thalamo-cortical synapses were used as input (see Preparation of thalamic input data, *Figure 4—figure supplement 4A1,A2*). Next, we used the region atlas (see Preparation of voxelized atlases) to find the corresponding depth bins in the model. This allowed the identification of all dendritic sections in the model whose center point fell within a depth bin. We then performed a random selection of those sections and placed synapses at random locations on each selected section, until the prescribed number of synapses for a given depth bin was reached (*Figure 4—figure supplement 4A2*). Sections were selected with probabilities proportional to their lengths and with replacement.

After all synapses for a thalamo-cortical projection had been placed, we mapped each of them to a presynaptic thalamic neuron. These neurons were not fully modeled, that is, they were not assigned a soma position or morphology, and the mapping simply allowed us to determine which synapses would be activated together. To parameterize the process, we used an estimate of the number of synapses formed by a single fiber and its horizontal spread (see Preparation of thalamic input data). We divided the total number of synapses placed ($590 \cdot 10^6$ core-type; $380 \cdot 10^6$ matrix-type) by the number per fiber to estimate the number of innervating fibers (approximately 100'000 for the 'core'-type projection; 73'000 'matrix'-type). These numbers were split between the eight subregions according to their relative volumes (see *Supplementary file 7*), with the following steps being executed separately for each of them.

We then abstractly modeled thalamo-cortical afferent axons as lines entering their respective subregion at the bottom of layer 6 with a certain horizontal reach for the formation of synapses.

This was done by first randomly distributing locations (one per fiber) across the boundary of layers 4 and 5 (*Figure 4—figure supplement 4B*) and then moving them 1500 µm along the negative voxel orientation (towards layer 6; *Figure 4—figure supplement 4B*, black dots). The resulting positions and orientation vectors were used as the starting position of the fibers and their directions, respectively (*Figure 4—figure supplement 4B*, black arrows). The presynaptic fiber of a synapse was then determined by a stochastic process based on the horizontal reach of individual fibers around their respective location (*Figure 4—figure supplement 4B*, red areas). This was parameterized as a Gaussian with a $\sigma = 120 \mu m$ for core-type, and $\sigma = 235 \mu m$ for matrix-type projections (see Preparation of thalamic input data). For each placed synapse, its distance to neighboring fibers was calculated and used as inputs into the Gaussian (*Figure 4—figure supplement 4D*). This distance was calculated as the distance between the location of the synapse and the line defined by the fiber's starting point and direction. The probability that any fiber was chosen as innervating fiber of the synapse was then proportional to the values (*Figure 4—figure supplement 4E*).

## Estimated volume of intrinsic mid-range axons

As the model did not contain mid-range axons, we estimated their volume based on mid-range synapse counts instead. We first counted the number of mid-range synapses on dendrites inside the volume of interest. We then converted the count into a volume by assuming that an axonal segment with a length of 5.4 µm and a diameter of 0.21 µm supported each synapse. The length was based on the inverse of the mean bouton density of excitatory axons in cortex (*Reimann et al., 2015*); the diameter was the mean diameter of axons in the model. As this excludes parts of the axon not forming boutons, this is a lower bound estimate.

## Measuring local connection probabilities

Local connection probabilities of the model were measured by emulating a multi-patch clamp sampling procedure as is often employed to measure connectivity in vitro. First, a pathway to sample, that is, pre- and post-synaptic neuron types, was selected and all neurons of non-participating types were discarded. Of the remaining neurons, one was randomly selected as an initial seed. Next, eleven additional neurons were selected by repeating the following procedure: Let $\mathcal{S} = s_0, s_1, ..., s_n$ be the set of already sampled neurons and $\mathcal{T}$ the remaining neurons. We calculate for all neurons $t_i \in \mathcal{T}$:

$$W_i = \prod_j \phi \left( \frac{D(t_i, s_j)}{\sigma} \right) \text{ if } D(t_i, s_j) > 10 \mu m \text{ else } 0, \tag{1}$$

where $\phi$ refers to the standard normal distribution and $D$ refers to the distance between two neuron somata in two dimensions. If the sampled pathway was intralaminar, the first dimension was orthogonal to the layer boundaries at the location of the first sampled neuron, and the second dimension was orthogonal to the first at a randomly chosen angle. If the sampled pathway was interlaminar, both dimensions were parallel to layer boundaries at the location of the first sampled neuron. The value of $\sigma$ was set to 38.82. Then, the next sampled neuron would be randomly picked from $\mathcal{T}$ with probabilities proportional to their respective $W_i$.

This roughly approximates the mainly two-dimensional sampling in slices of brain tissue while remaining comparable between intra- and interlaminar pathways. The value of $\sigma$ was determined to yield pairs at distances below 100 µm with high probability. Specifically, this puts $D = 100 \mu m$ at around 2.5 standard deviations or the 95th percentile of the normal distribution used.

## Literature sources for connection probabilities

We used the data collected in 'Figure 4-1, XLSX file' of *Zhang et al., 2019*. Reported connection probability values were classified into a pathway as follows. An initial classification of source and target into excitatory or inhibitory was extracted from the column 'Connection type'. Next, source and target layers were determined based on the columns 'Presynaptic Type' and 'Postsynaptic Type'. This was straightforward, as the values were consistently formatted to list the layer first, followed by additional specification, e.g. 'L4 FS IN'. Inhibitory types were further broken down: If 'Presynaptic Type' or 'Postsynaptic Type' specified one of 'PV' or 'BC' it was classified as 'PV'; if it specified one of 'SOM', 'MC' or 'DBC' it was classified as 'SST'; otherwise it was classified as 'INH'.

For *Figure 4—figure supplement 2B* we used all sources where the value of 'Species' was either 'rat' or 'mouse'. For *Figure 4—figure supplement 2C* we additionally ensured that the value of 'Age' was prescribed in postnatal days and the reported interval contained P14. For *Figure 4—figure supplement 2D* only sources where the value of 'Species' was 'rat' were used.

## Connection probability of L5_PCs in the MICrONS data

We analyzed the 'minnie65_public_v117' release of the MICrONS data (https://www.microns-explorer.org/cortical-mm3). We restricted the analysis to neurons where the proofreading status was listed as 'clean' or 'extended'. We then considered neurons where the type of the 'allen soma coarse cell class model' was listed as '5 P_IT', '5 P_PT' or '5 P_NP', resulting in 37 neurons. In that group, 154 pairs were within 100 µm. Of those, 13 were connected according to the 'synapses_pni_2' table of the data release.

## Validation of synaptic connectivity

The part of the connectome derived with a apposition-based approach (local connectivity) was constrained by anatomical data on bouton densities and mean numbers of synapses per connection for different morphological types (*Reimann et al., 2015*). As such, we validated that the results match these data (*Figure 4—figure supplement 3A,B*). As previously found, the only mismatch lied in the emerging bouton densities of Chandelier Cells. These neurons form synapses only onto the axon initial segment of other neurons, which we model by disregarding appositions on the dendrites. For two Chandelier types, this resulted in an insufficient number of appositions to fulfill bouton density constraints (*Figure 4—figure supplement 3A*, black arrow).

Arguably the most important constraint is the number or density of excitatory synapses in pathways between individual subregions, as it determines the overall excitability of the model and the velocity of the spread of activity. Here, we compared the total number of excitatory synapses, that is, the union of the output of both algorithms to the data (*Figure 4—figure supplement 3C*), finding a robust qualitative match. Due to the stochastic nature of the connectivity algorithms, an exact match cannot be expected.

Finally, the mid-range connectivity algorithm was further constrained by predicted topographical mapping between regions (*Figure 4—figure supplement 3D*) and synapse layer profiles (Modeling synaptic connectivity), which we validated against the data.

### Validation of thalamo-cortical innervation

In silico synapse density profiles of VPM and POm projections were validated against the in vitro layer profiles from *Meyer et al., 2010* that were used as the input recipes (*Figure 6—figure supplement 2A,B*). To account for non-uniform thicknesses across regions, the depth values of the density profiles were normalized to the maximum thickness of each of the eight subregions. There is a decent match between the recipe and the actual layer profiles, but with a 15% overshoot at peak densities. This can be explained by the fact that the numbers of synapses to be placed were computed based on region bounding boxes which were larger than the actual volumes. Therefore, the actual densities are slightly higher when computing them on a voxel basis as it was done in this validation.

### Calculation of common thalamic innervation

For a given thalamo-cortical pathway we calculated the common thalamic innervation (CTI) as follows for all neurons: First, for each neuron the set of thalamic fibers innervating it with at least one synapse was identified. Next, we iterated over all pairs of neurons in the model, comparing their sets of innervating fibers. Specifically, we calculated the size of the intersection of the sets and divided it by the size of their union. The resulting CTI was a measure between 0 (no overlap) and 1 (complete overlap).

### Generation of columnar subvolumes

We partitioned the modeled volume into subvolumes with shapes approximating hexagonal prisms. To that end, we represented each voxel by the flattened location of its center. In the resulting two-dimensional coordinate system, we distributed seed points in a hexagonal grid with a distance of 460 µm (based on the 230 µm column radius of *Markram et al., 2015*) between neighboring points. Next, for each voxel we determined to which seed point it was closest. All voxels that were closest to the same seed point were then grouped together as parts of the same subvolume. As subvolumes near the periphery of the model could be incomplete, that is, not form complete hexagonal prisms, we excluded them from further analysis if either of the following two conditions was met: (1) The subvolume had fewer than six neighbors in the hexagonal grid; (2) Its volume was less than 0.166 mm$^3$. That threshold is based on the volume of a circular prism with a radius of 230 µm and a height of 1000 µm.

### Columnar volumes and their conicality

Columnar volumes were defined in a flat view of the model, generated as in *Bolaños-Puchet et al., 2024*; *Bolaños-Puchet et al., 2023*. In the two-dimensional coordinate system, we defined a hexagonal grid with a radius (large diagonal of a hexagon) as indicated in the text. Then, each voxel (38.7348 µm resolution) was associated with the hexagon that contained the flat coordinates of its central point. Thus, each hexagon defined a continuous columnar volume that sampled all cortical layers. To calculate its conicality, we first found its central vertical axis (orthogonal to layer boundaries). Then we conducted a linear fit of cortical depth against distance from the vertical axis of all voxels contained in a column. The slope of the linear fit was defined as the conicality measure.

### Generation of random connectivity control models

This section contains definitions of the random connectivity control models and explains how these were generated, presented in weakly increasing order of complexity.

The *Erdős–Rényi* controls (ER) are random directed networks where the edges are added with a fixed probability independently at random. The controls were constructed by taking each ordered pair of nodes $(i, j)$ and adding an edge from $i$ to $j$ at random with probability $p = \frac{E}{N(N-1)}$, where $E$ (respectively $N$) is the number of edges (respectively nodes) in the original network.

The *stochastic block model* controls (SBM) are random networks where the edges are added independently at random with a fixed probability dependent on the m-type pathway of the pre- and post-synaptic neurons. These controls were built as for the ER-controls, but with a different probability for each pathway. More precisely, if $(i, j)$ are two neurons of m-types $A$ and $B$, then their probability of connection is

$$p_{AB} = \begin{cases} \dfrac{E_{AB}}{(N_{AB})^2} & \text{if } A \neq B, \\ \dfrac{E_{AA}}{(N_{AA})(N_{AA} - 1)} & \text{if } A = B, \end{cases}$$

where $N_{AB}$ is the number of neurons of m-type $A$ and $B$ and $E_{AB}$ is the number of edges from neurons of type $A$ to neurons of type $B$ in the original network.

The *directed configuration model* controls (CM) are random networks that closely approximate a given in/out-degree sequence, that is, a vector of length the number of nodes, whose entries are their in/out-degrees. To build these controls, we encode the edges of the original matrix by two vectors, *sources* and *targets*, such that $(sources[i], targets[i])$ is a directed edge of the matrix (this corresponds to a binary matrix in coordinate format, useful when working with sparse matrices). Then, we shuffle the entries of both vectors *sources* and *targets* independently, which gives a new directed network, with the same degree sequence as the original. This new network might contain loops or parallel edges, so we remove them, thus this construction only approximates the original degree sequence. The density of loops and parallel edges tends to decrease as the number of nodes increases (**Newman, 2003**), so the approximation is good. Indeed, in our controls for local connectivity, $235,574.0 \pm 376.0$ (mean $\pm$ std) connections out of $2,050,028,490$ were missing, which is less than 0.012% of all connections. For mid-range connectivity the number was $629,493.4 \pm 2,228.0$ out of $2,482,296,102$, which is less that 0.026% of all connections.

Finally, the distance block model controls (DBM) are random networks where the edges are added independently at random with a probability $p_A(d) = \alpha_A \cdot e^{-\beta_A \cdot d}$, that is, exponentially decreasing with distance $d$, where $\alpha_A$ is the probability at distance zero and $\beta_A$ the decay constant, both depending only on the pre-synaptic cell m-type $A$. For local connectivity, the distance between a pair of neurons used was the Euclidean distance of their position in space (in µm). For mid-range connectivity, the specific method used to originally construct the connectome (**Reimann et al., 2019**) was taken into account, preferentially connecting neurons at one location within a subregion $A$ to neurons in another subregion $B$ by first parameterizing linear transformations of the flattened coordinates of the two regions. The resulting *virtual coordinate system* $V_{A,B}$ is then used to connect pairs of neurons from $A$ and $B$ in a distance-dependent way. Therefore, the distance considered in this control between pairs of neurons was the Euclidean distance between these virtual coordinates.

For computational reasons, the model coefficients $\alpha_A$ and $\beta_A$ were estimated by randomly subsampling (up to) 100000 pre-synaptic neurons of m-type $A$ and 100000 post-synaptic neurons of any type and computing their pairwise distances. For local connectivity, distances from 0 to 1000 µm were then divided into 20 evenly spaced 50 µm bins. For mid-range connectivity, distances from 0 to 400(a.u.) were divided into 20 evenly spaced 20 (a.u) bins. Connection probabilities were estimated in each bin by dividing the number of existing connections by the number of possible connections between all pairs of neurons within that bin (excluding connections at distance zero, that is, between one neuron with itself). Model coefficients were then determined by fitting the exponential probability function $p(d)$ to these data points. The whole procedure was repeated ten times with different random sub-samples of neurons, and the averaged model coefficients over these ten estimates were used to build the controls. Overall, the relative standard errors of the mean over the ten estimates were on average across all m-types less than 1%. For local connectivity, model coefficients of all 60 pre-synaptic m-types were found to be on average $\overline{\alpha} = 0.138 \pm 0.102 \sim (SD)$ and $\overline{\beta} = 0.0096 \pm 0.002 \sim (SD)$ respectively. For mid-range connectivity, the model coefficients of all 18 m-types that had any outgoing mid-range connections were on average $\overline{\alpha} = 0.0006 \pm 0.001 \sim (SD)$ and $\overline{\beta} = 0.012 \pm 0.001 \sim (SD)$ respectively.

## Modularity

The *modularity* of a network is a metric that determines to which extent it is structured by modules or communities. Intuitively, a network has high modularity if nodes within the modules are densely interconnected, while nodes of different modules are sparsely interconnected.

Formally, consider a directed network with no loops or double edges. It can be represented by a binary matrix $A$ with zeros in the diagonal where the entry $A_{ij}$ is 1 if there is an edge from $i$ to $j$ and 0 otherwise. The *modularity of a partition $\mathcal{C}$* of the nodes of the network into modules is given by:

$$\mathcal{Q}_\mathcal{C} = \sum_{i,j} \left( \frac{A_{ij}}{m} - P_{ij} \right) \delta(c_i, c_j),$$

where $m$ is the number of edges that is, the number of non-zero entries in $A$, $P_{ij}$ is the probability of having an edge from $i$ to $j$ in a directed configuration model of $A$, and

$$\delta(c_i, c_j) = \begin{cases} 1 & \text{if } i \text{ and } j \text{ are in the same module,} \\ 0 & \text{otherwise.} \end{cases}$$

Thus, the function $\mathcal{Q}_\mathcal{C}$ contrasts the existence of an edge between two nodes in the same module with the probability that it would appear by chance. This probability is given by

$$P_{ij} = \frac{d_i^{out} d_j^{in}}{m^2},$$

where $d_i^{out}, d_j^{in}$ are the in- and out-degrees of the nodes $i$ and $j$ respectively. The *modularity of the network $\mathcal{Q}_A$* is the maximal value attained by $\mathcal{Q}_\mathcal{C}$ across all possible partitions of the nodes of $A$. See *Nicosia et al., 2009* for further details. It has been shown that computing the modularity of a network is NP-complete (*Brandes et al., 2007*), thus only heuristic approximations are possible. To approximate the modularity of each of the columns in the parcellation described in *Figure 7*, we used the Louvain algorithm for directed networks (*Dugué and Perez, 2015*) implemented in Scikit-network (*Bonald et al., 2020*). To control for the effect of size and density of the columns on their modularity we also computed the modularity of corresponding ER-controls.

Note that the output of the Louvain algorithm is not deterministic. To further asses the its effect on the modularity approximation we focused on the columns with high modularity values that is, those with modularity values higher than 0.4. For each of these columns, we recomputed the modularity of a subnetwork obtained by subsampling 50% of the nodes at random. We computed the modularity of the subnetwork with respect to two partitions: First, where the modules are inherited from the modules determined by the Louvain algorithm in the full column, we called these the *original labels*. Second, by re-running the Louvain algorithm from scratch on the subcolumn and thus determining new modules of the subnetwork, we called these the *new labels*. We repeated this procedure for 50 different random samples for each column and contrasted the distribution of the modularity values obtained in both modalities as well as those obtained for the full column.

## Node participation and simplicial core

For any given node in a directed network, its *node participation* is the number of simplices that it is part of. This value can be further split by dimension, giving rise to the notion of $n$-*node participation*. Given that any directed simplex has a single source and a single target, node participation can be further refined to participation as a source or participation as a sink. For $n = 1$, source node participation is equivalent to out-degree, sink node participation is equivalent to in-degree and node participation is equivalent to the total degree (i.e., the sum of in- and out-degrees). Thus, node participation can be thought of as a higher dimensional version of degree. The $n$-*simplicial core* of a network is the sub-network on the nodes that participate in simplices of dimension $n$ or higher, or equivalently on the nodes whose $n$-node participation is greater than 0. The *simplicial core* is the sub-network on the nodes that participate in simplices of maximal dimension. These concepts are a generalization of the notion of core of a network, which is a notion that is solely based on degree, by taking into account higher order interactions. We refer to the ($n$-) simplicial core of the local/mid-range network as the ($n$-)*local/mid-range core*, and we refer to the union of both as *the core*.

## Finding and counting simplices in connectivity networks

Directed simplices were computed using the custom-made C++ package flagser-count. This code is a variation of the flagser package (*Lütgehetmann et al., 2020*). The code takes as input the adjacency matrix of a directed network in compressed sparse row (or column) format, outputs the number of directed cliques in the network, and includes the option to print all simplices. There is also the option to output node participation, so for every node $v$ and every dimension $d$ a value is given for the number of $d$-simplices that contain $v$. The algorithm considers each node independently as a source node and then performs a depth-first search on that node to find all simplices, where at each step it creates a new simplex by adding any node that is an out-neighbor of all nodes in the current simplex.

Since each source node is considered independently, the computations can be easily parallelized. The simplex count computations were conducted on the Blue Brain high performance computing system. Each computation was run on two Intel Xeon Gold 6248 CPUs using a total of 40 physical cores. To compute the simplex counts for the local network took 3 hours and 15 minutes using 8.2 GB of memory, the mid-range network took 14 hours and 44 minutes using 9.8 GB of memory and the combined network took 115 days and 20 hours using 69.5 GB of memory. The transpose adjacency matrices of the mid-range and combined networks were used, which have the same simplex counts, because these were significantly faster to compute due to the degree distributions. The computation for the combined network was attempted on the non-transposed matrix and ran for 121 days, and computed partial counts, but encountered 898 nodes for which the counts could not be computed within 24 hours, such nodes were deemed too computational expensive and skipped.

## Rich-club and generalized rich-club

The *k-rich-club coefficient* of an undirected network is the density of the subnetwork on the nodes of degree greater than $k$, and is given by the formula

$$\phi(k) = \frac{2E_{>k}}{N_{>k}(N_{>k} - 1)},$$

where $N_{>k}$ is the number of nodes of degree greater than $k$, and $E_{>k}$ is the number of edges between the nodes in $N_{>k}$. We call the function $\phi$ the *rich-club curve*. For a directed network we obtain three variations of this curve depending on whether we consider in-degree, out-degree, or total-degree (the sum of in- and out-degree), and we remove the coefficient 2 due to the fact that the maximum number of edges is now $N(N-1)$ instead of $N(N-1)/2$. This gives us three formulas

$$\phi^{total}(k) = \frac{E_{>k}^{total}}{N_{>k}^{total}(N_{>k}^{total} - 1)}, \quad \phi^{in}(k) = \frac{E_{>k}^{in}}{N_{>k}^{in}(N_{>k}^{in} - 1)}, \quad \phi^{out}(k) = \frac{E_{>k}^{out}}{N_{>k}^{out}(N_{>k}^{out} - 1)}, \tag{2}$$

where $N_{>k}^{total}$ (resp. $E_{>k}^{total}$) is equivalent to $N_{>k}$ (resp. $E_{>k}$) but restricted to the total degree, and similarly for in and out.

The rich-club coefficient can be naturally generalized by replacing the degree with any network metric (*Opsahl et al., 2008*). In particular, we define the *simplicial rich-club coefficient* by

$$\phi^d(k) = \frac{E_{>k}^d}{N_{>k}^d(N_{>k}^d - 1)}, \tag{3}$$

where $N_{>k}^d$ is the number of nodes with $d$-node participation at least $k$, and $E_{>k}^d$ is the number of edges between them. Note that $\phi^1 = \phi^{total}$ since a 1-simplex is simply an edge of the network.

A large rich-club coefficient value is said to indicate a preference of high degree nodes to connect together. However, it is shown in *Colizza et al., 2006* that the rich-club coefficient is increasing even in Erdős–Rényi networks, and is actually a consequence of higher degree nodes naturally being more likely to connect. Therefore, it is important to normalize the rich-club coefficient by dividing by the rich-club of an appropriate control, that is, a control with the same degree distribution such as a configuration model, see Generation of random connectivity control models. Hence we define the *normalized rich-club coefficient* as

$$\rho^{total}(k) = \frac{\phi^{total}(k)}{\phi_{rand}^{total}(k)}, \tag{4}$$

where $\phi_{rand}^{total}(k)$ is the rich-club coefficient of a configuration model control of the original network, and we similarly define $\rho^{in}(k)$ and $\rho^{out}(k)$. A network is said to have a *rich-club effect* if $\rho^{total}(k) > 1$, as this indicates the high degree nodes connect together more than is 'expected'. The importance of this normalization can be seen in **Figure 8—figure supplement 1**, where B1 seems to indicate the presence of a rich-club effect for the mid-range network, which disappears when normalized in B2.

Normalizing the simplicial rich-club is difficult as it requires a control model on directed flag complexes that fixes node participation. Randomly sampling simplicial complexes with fixed simplex counts or other high dimensional network properties is an active area of research (**Kahle, 2014**; **Young et al., 2017**; **Unger and Krebs, 2024**), but as of yet there is no known appropriate control model for the simplicial rich-club, and it is a topic that warrants further investigation.

## Paths and path distances

Let $G$ be a directed network on $n$ nodes, with adjacency matrix $A$. A *path* of length $k$ (or $k$-path) from node $u$ to node $v$ is a sequence of nodes $u = x_1, \ldots, x_{k+1} = v$ such that $(x_i, x_{i+1})$ is an edge of $G$ for all $i = 1, \ldots, k$. The *path distance* between $u$ and $v$ is the length of the shortest path from $u$ to $v$. A path from $u$ to $v$ is *geodesic* if its length is the path distance from $u$ to $v$.

The number of $k$-paths between all nodes in $G$ can be easily computed using matrix multiplication, since the number of $k$-paths between $u$ and $v$ is given by $A_{u,v}^k$, where $A$ is the adjacency matrix of $G$ (**Diestel, 2005**). Consequently, the number of $k$-paths from the nodes $i_1, \ldots, i_t$ to all other nodes in $G$ is given by

$$A_{[i_1,\ldots,i_t]}^k = A_{[i_1,\ldots,i_t]}A^{k-1} = A_{[i_1,\ldots,i_t]}^{k-1}A,$$

where $A_{[i_1,\ldots,i_t]}$ is the $t \times n$ matrix obtained from $A$ by taking only the rows $i_1, \ldots, i_t$. Note that the smallest $k$ for which entry $i,j$ is nonzero is the path distance from $i$ to $j$.

Using this approach, we were able to compute the path distances in the combined network between pairs of all 396 nodes in the local and mid-range cores. We ran the computation in parallel by partitioning the nodes into 13 sets $C_1, \ldots, C_{13}$ of approximately 23 nodes each and computing

$$A_{[C_i]}^k = (\ldots(A_{[C_i]}\underbrace{A)A\ldots A}_{\times k-1}).$$

Note that the order of the brackets ensures that at each step a $23 \times 4234929$ matrix is computed, rather than a $4234929 \times 4234929$ matrix.

Once we had the path distances between the nodes in the local and mid-range cores, we computed the geodesic paths between them. This was done using the custom C++ package pathfinder. This code functions in a similar way to flagser-count, except that at each step it creates a new path by considering any node that is an out-neighbor of the final node in the current path. At input the start-nodes, end-nodes, and path distances can be specified. To compute all geodesic paths of length at most three between nodes within the local and mid-range cores took 8 hr and 47 min, using 51.9 GB of memory. Between these 396 nodes there are 6,383 geodesic 1-paths, 1,039,814 geodesic 2-paths, and 21,701,345 geodesic 3-paths. To compute all geodesic 4-paths was attempted, but after running for 24 hr failed to finish, and is likely to be computationally infeasible due to the exponential growth of the number of paths.

## Generation of EM-like volumes

We emulated the specificities of the electron microscopic studies of **Motta et al., 2019** and **Schneider-Mizell et al., 2024** as follows. First, in **Motta et al., 2019** the fragments of axons and dendrites inside a volume are reconstructed as well as their synaptic connections. Then, axon and dendrite fragments with at least 10 synapses inside the volume are considered for analysis. In the model, we began by loading the three-dimensional locations of all synapses between neurons in a 100 µm columnar volume. Next, we determined which synapses are contained in a box-shaped subvolume at the top of layer 4 of the model. The size of the box was determined to contain the same number of synapses

(approximately 145,000) as the volume of *Motta et al., 2019*. This was achieved at $110x110x85\mu m$, about twice as large as the reference volume, as it does not contain synapses extrinsic to the model (also we do not know to what degree the dimensions of the reference volume were corrected for shrinkage). We then determined the types of postsynaptic compartments. Synapses onto somata and axon initial segments were labeled as such; of the rest, if the soma of the targeted neuron was inside the box volume, it was tagged as proximal dendrite; of the rest, if the targeted neuron was inhibitory, it was tagged as smooth dendrite; remaining synapses onto apical dendrites were tagged as such; all others were tagged 'other'. We applied the same filtering as the reference, removing axons, dendrites (and their synapses) with less than 10 synaptic contacts.

On the other hand, *Schneider-Mizell et al., 2024* determined which neurons had their somata inside a volume, and then reconstructed their entire dendritic and axonal trees, even parts outside the volume. The volume sampled all layers and was approximately $100x100\mu m$ wide, but was not exactly box-shaped, as it followed the main direction of contained apical dendrites. We emulated this by defining a subvolume in flat mapped coordinates (*Bolaños-Puchet et al., 2024*; *Bolaños-Puchet et al., 2023*): We considered all neurons whose flattened coordinates were inside a $100x100\mu m$ square. As the flat map flattens away the vertical dimension along which apical dendrites are oriented in the model, this corresponded to the the sampling of *Schneider-Mizell et al., 2024*. Then, all synaptic connections between the sampled neurons were analyzed as in the reference.

### Binomial 'first hit' model

Control models to compare the distributions of postsynaptic compartments against were generated as described in the reference (*Motta et al., 2019*). The binomial model assumes that each synapse of an axon is formed with a fixed probability $p_{type}$ onto a given postsynaptic compartment type, hence the expected distribution of the number of synapses onto that type is a binomial one. A binomial model is fitted for each combination of axon type (excitatory vs. inhibitory) and postsynaptic compartment type as follows. For all axon fragments of the given type, we consider the total number of synapses formed and whether it forms at least one synapse onto the given compartment type. For a given axon fragment, the likelihood of the observation is $(1 - p_{type})^{n_{axon}}$ if no synapse onto the type is observed and $1 - (1 - p_{type})^{n_{axon}}$ otherwise, where $n_{axon}$ is the number of synapses of the axon fragment in the volume. Then, $p_{type}$ is estimated by maximizing the joint likelihood of all observations, that is, over all axons of the given type.

## Acknowledgements

The authors would like to thank Giuseppe Chindemi, Javier DeFelipe and Rajnish Ranjan for help with the scientific development of the model; Tristan Carel, James Dynes, Stefan Eilemann, Bruno Magalhães, Juan Hernando Vieites and Arseny Povolotsky for contributions and support to the engineering challenges; the BBP Core Services team for responding to IT requests and services surrounding the research; Marwan Abdellah, Elvis Boci and Nadir Roman Guerrero for help with visualizations of the model; Zoltán Kisvárday for supervision of morphology reconstruction efforts; Eva Kenny, Silvia Scarabelli and Riccardo Sinsi for help with project management; and Karin Holm, Akiko Sato and Georges Khazen for support of manuscript development and helpful discussions. Funding This study was supported by funding to the Blue Brain Project, a research center of the École polytechnique fédérale de Lausanne (EPFL), from the Swiss government's ETH Board of the Swiss Federal Institutes of Technology. RL, JPS and JL were supported by EPSRC under grant number EP/P025072/1. RL was supported by a collaboration grant from EPFL. S.R. is supported by a Marie Skłodowska-Curie Global Fellowship Agreement 842492; Newcastle University Academic Track (NUAcT) Fellowship; Fulbright Research Scholarship; Lister Institute Prize Fellowship; Academy of Medical Sciences Springboard Award; Research grant from the Air Force Office of Scientific Research (FA9550-23-1-0533); International Brain Research Organization (IBRO) Early-career Award; Theoretical Sciences Visiting Program (TSVP) at the Okinawa Institute of Science and Technology (OIST).

# Additional information

## Competing interests

Eilif B Muller: Reviewing editor, eLife. The other authors declare that no competing interests exist.

## Funding

| Funder | Grant reference number | Author |
|---|---|---|
| Engineering and Physical Sciences Research Council | EP/P025072/1 | Jānis Lazovskis<br>Jason P Smith<br>Ran Levi |
| École Polytechnique Fédérale de Lausanne | collaboration grant | Ran Levi |
| HORIZON EUROPE Marie Sklodowska-Curie Actions | 842492 | Srikanth Ramaswamy |
| Newcastle University | Academic Track (NUAcT) Fellowship | Srikanth Ramaswamy |
| Fulbright Association | Research Scholarship | Srikanth Ramaswamy |
| Lister Institute of Preventive Medicine | Prize Fellowship | Srikanth Ramaswamy |
| Academy of Medical Sciences | Springboard Award | Srikanth Ramaswamy |
| Air Force Office of Scientific Research | FA9550-23-1-0533 | Srikanth Ramaswamy |
| International Brain Research Organization | Early-career Award | Srikanth Ramaswamy |
| Okinawa Institute of Science and Technology Graduate University | Theoretical Sciences Visiting Program | Srikanth Ramaswamy |

| Funder | Grant reference number | Author |
|---|---|---|
| ETH Board of the Swiss Federal Institutes of Technology | Funding to the Blue Brain Project | Michael W Reimann<br>Sirio Bolaños-Puchet<br>Jean-Denis Courcol<br>Alexis Arnaudon<br>Benoît Coste<br>Fabien Delalondre<br>Thomas Delemontex<br>Adrien Devresse<br>Hugo Dictus<br>Alexander Dietz<br>András Ecker<br>Cyrille Favreau<br>Gianluca Ficarelli<br>Mike Gevaert<br>Joni Herttuainen<br>James B Isbister<br>Lida Kanari<br>Daniel Keller<br>James King<br>Pramod Kumbhar<br>Samuel Lapere<br>Huanxiang Lu<br>Fernando Pereira<br>Judit Planas<br>Christoph Pokorny<br>Juan Luis Riquelme<br>Armando Romani<br>Ying Shi<br>Vishal Sood<br>Werner Van Geit<br>Liesbeth Vanherpe<br>Matthias Wolf<br>Felix Schürmann<br>Eilif B Muller<br>Henry Markram<br>Srikanth Ramaswamy |

The funders had no role in study design, data collection and interpretation, or the decision to submit the work for publication.

## Author contributions

Michael W Reimann, Conceptualization, Software, Formal analysis, Supervision, Validation, Investigation, Visualization, Methodology, Writing – original draft, Project administration, Writing – review and editing; Sirio Bolaños-Puchet, Data curation, Formal analysis, Methodology, Writing – review and editing; Jean-Denis Courcol, James King, Resources, Software, Supervision; Daniela Egas Santander, Formal analysis, Investigation, Visualization, Methodology, Writing – original draft; Alexis Arnaudon, Software, Writing – review and editing; Benoît Coste, Fabien Delalondre, Thomas Delemontex, Adrien Devresse, Gianluca Ficarelli, Joni Herttuainen, Juan Luis Riquelme, Liesbeth Vanherpe, Matthias Wolf, Software; Hugo Dictus, Mike Gevaert, Vishal Sood, Software, Validation; Alexander Dietz, Validation; András Ecker, Software, Formal analysis, Validation, Visualization, Writing – review and editing; Cyrille Favreau, Visualization; James B Isbister, Writing – review and editing; Lida Kanari, Data curation, Formal analysis, Validation, Visualization, Methodology, Writing – original draft; Daniel Keller, Data curation, Methodology, Writing – original draft; Pramod Kumbhar, Fernando Pereira, Resources, Software; Samuel Lapere, Judit Planas, Project administration; Jānis Lazovskis, Software, Visualization, Methodology; Huanxiang Lu, Data curation, Software; Nicolas Ninin, Methodology; Christoph Pokorny, Validation, Visualization, Writing – original draft; Armando Romani, Supervision, Methodology, Project administration; Ying Shi, Data curation; Jason P Smith, Software, Investigation, Methodology, Writing – original draft; Mohit Srivastava, Investigation, Writing – original draft; Werner Van Geit, Validation, Methodology; Ran Levi, Kathryn Hess, Supervision, Methodology, Project administration, Writing – review and editing; Felix Schürmann, Resources, Supervision, Project administration; Eilif B Muller, Conceptualization; Henry Markram, Conceptualization, Supervision, Funding acquisition, Project administration; Srikanth Ramaswamy, Conceptualization, Supervision, Methodology, Writing – original draft, Project administration, Writing – review and editing

## Author ORCIDs

Michael W Reimann https://orcid.org/0000-0003-3455-2367
Sirio Bolaños-Puchet https://orcid.org/0000-0003-4049-6488
Daniela Egas Santander https://orcid.org/0000-0001-9838-7992
András Ecker https://orcid.org/0000-0001-9635-4169
Judit Planas https://orcid.org/0000-0002-8221-7988
Eilif B Muller https://orcid.org/0000-0003-4309-8266

## Ethics

Approval of Local Ethics Committee for Animal Research Studies at the University of Debrecen in line with European Union guidelines for the care of laboratory animals, Directive 2010/63/EU.

Reviewer #1 (Public review): https://doi.org/10.7554/eLife.99688.3.sa1
Reviewer #2 (Public review): https://doi.org/10.7554/eLife.99688.3.sa2
Reviewer #3 (Public review): https://doi.org/10.7554/eLife.99688.3.sa3
Author response https://doi.org/10.7554/eLife.99688.3.sa4

# Additional files

## Supplementary files

Supplementary file 1. Descriptions of indvidual validations and their results.

Supplementary file 2. List of assumptions made in model building.

Supplementary file 4. List of predictions made and experiments to test them.

Supplementary file 5. Normalized depths used for layer boundaries.

Supplementary file 6. Placement features used for modeling. The right column denotes the vertical target interval by specifying a layer and a relative offset within the layer, with 0.0 indication the bottom and 1.0 the top of the layer.

Supplementary file 7. Number of thalamic fibers providing inputs to the model and each of its subregions.

Supplementary file 9. Morphological types (m-types) used in the model.

MDAR checklist

## Data availability

Data on cell densities, bouton densities and mean numbers of synapses per connection were reported in original publications. Data on synapse densities of thalamo-cortical inputs and their projection axon widths are reported in figures and text in this publication. Volumetric atlases, neuron reconstructions and the description of the model in SONATA format have been deposited at Zenodo or Harvard Dataverse and are publicly available as of the date of the publication. Deposited models include reconstructed neuron morphologies. DOIs of these datasets are listed below in the list of newly generated datasets. Original code has been deposited at Zenodo or github and is publicly available as of the date of publication. DOIs or links are listed in the key resources table. Any additional information required to reanalyze the data reported in this paper is available from the lead contact upon request: Michael W. Reimann (mwr@reimann.science).

The following datasets were generated:

| Author(s) | Year | Dataset title | Dataset URL | Database and Identifier |
|---|---|---|---|---|
| Reimann MW | 2023 | A Model of Rat Non-barrel Somatosensory Cortex Anatomy | https://doi.org/10.5281/zenodo.8155899 | Zenodo, 10.5281/zenodo.8155899 |
| Reimann MW | 2024 | Brain atlas data for the BBP Somatosensory Cortex model | https://doi.org/10.7910/DVN/QREN2T | Harvard Dataverse, 10.7910/DVN/QREN2T |

*Continued*

| Author(s) | Year | Dataset title | Dataset URL | Database and Identifier |
|---|---|---|---|---|
| Reimann MW | 2024 | BBP Somatosensory Cortex model - SONATA version | https://doi.org/10.7910/DVN/HISHXN | Harvard Dataverse, 10.7910/DVN/HISHXN |
| Pokorny C | 2024 | Rewired connectome of an SSCX model with inhibitory targeting based on trends found in MICrONS data | https://doi.org/10.5281/zenodo.10677883 | Zenodo, 10.5281/zenodo.10677883 |
| Isbister JB | 2024 | A Model of Rodent Neocortical Micro- and Mesocircuitry | https://doi.org/10.5281/zenodo.11113043 | Zenodo, 10.5281/zenodo.11113043 |

The following previously published dataset was used:

| Author(s) | Year | Dataset title | Dataset URL | Database and Identifier |
|---|---|---|---|---|
| Bolanos-Puchet S | 2024 | Enhanced atlases and flatmaps of rodent neocortex | https://doi.org/10.5281/zenodo.11218079 | Zenodo, 10.5281/zenodo.11218079 |

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
