## [Editor Report · eLife Assessment]

This manuscript reports a detailed model of juvenile rat somatosensory cortex, consisting of 4.2 million morphologically and biophysically detailed neuron models, arranged in space and connected according to diverse experimental data - a **valuable** tool for the field. The construction of the model is based on a methodology with **solid** supporting evidence. It should be noted that, by necessity, such a large-scale model development involves many assumptions, interpolations, and decisions that could have compounding downstream effects on further analyses that may be difficult to disambiguate.

---

## [Referee Report · Reviewer #1 (Public review)]

Summary:

In this study, the authors describe the construction of an extremely large-scale anatomical model of juvenile rat somatosensory cortex (excluding the barrel region), which extends earlier iterations of these models by expanding across multiple interconnected cortical areas. The models are constructed in a way to maintain biological detail from a granular scale - for example, individual cell morphologies are maintained, and synaptic connectivity is founded on anatomical contacts. The authors use this model to investigate a variety of properties, from cell-type specific targeting (where the model results are compared to findings from recent large-scale electron microscopy studies) to network metrics. The model is also intended to serve as a platform and resource for the community by being a foundation for simulations of neuronal circuit activity and for additional anatomical studies that rely on the detailed knowledge of cellular identity and connectivity.

Strengths:

As the authors point out, the combination of scale and granularity of their model are what make this study valuable and unique. The comparisons with recent electron microscopy findings are some of the most compelling results presented in the study, showing that certain connectivity patterns can arise directly from the anatomical configuration, while other discrepancies highlight where more selective targeting rules (perhaps based on molecular cues) are likely employed. They also describe intriguing effects of cortical thickness and curvature on circuit connectivity and characterize the magnitude of those effects on different cortical layers.

The detailed construction of the model is drawn on wide range of data sources (cellular and synaptic density measures, neuronal morphologies, cellular composition measures, brain geometry, etc.) that are integrated together; other data sources are used for comparison and validation. This consolidation and comparison also represents a valuable contribution to the overall understanding of the modeled system.

Weaknesses:

The scale of the model, which is a primary strength, also can carry some drawbacks. In order to integrate all the diverse data sources together, many specific decisions must be made about, for example, translating findings from different species or regions to the modeled system, or deciding which aspects of the system can be assumed to be same and which should vary. All these decisions will have effects on the predicted results from the model, which could limit the types of conclusions that can be made (both by the others and by others in the community who may wish to use the model for their own work). However, the public release of the models and most of the associated tools does provide others a somewhat easier path to modify and evaluate this iteration of the model for their own studies.

Overall, the model presented in this study represents an enormous amount of work and stands as the basis for other work by the same group as well as a unique resource for the community, even while acknowledging that it may be somewhat unwieldy for the community to employ due to the weight of its manifold specific construction decisions, size, and complexity.

---

## [Referee Report · Reviewer #2 (Public review)]

Summary:

The authors build a colossal anatomical model of juvenile rat non-barrel primary somatosensory cortex, including inputs from the thalamus. This enhances past models by incorporating information on the shape of the cortex and estimated densities of various types of excitatory and inhibitory neuron across layers. This is intended to enable analysis of the micro- and mesoscopic organisation of cortical connectivity and to be a base anatomical model for large-scale simulations of physiology.

Strengths:

• The authors incorporate many diverse data sources on morphology and connectivity.

• This paper takes on the challenging task of linking micro- and meso-scale connectivity

• By building in the shape of the cortex, the authors were able to link cortical geometry to connectivity. In particular they make an unexpected prediction that cortical conicality affects the modularity of local connectivity, which should be testable.

• The author's analysis of the model led to the interesting prediction that layer 5 neurons' connect local modules, which may be testable in the future, and provide a basis to link from detailed anatomy to functional computations.

• The visualisation of the anatomy in various forms is excellent

• The model is openly shared

Weaknesses:

• There is no effort to determine how specific or generalisable the findings here are to other parts of cortex.

• Although there is a link to physiological modelling in another paper, there is no clear pathway to go from this type of model to understanding how the specific function of the modelled areas may emerge here (and not in other cortical areas).

• Some of the decisions seem a little ad-hoc, and the means to assess those decisions is not always easily available to the reader

• The shape of the juvenile cortex - a key novelty of this work - was based on merely a scalar reduction of the adult cortex. This is very surprising, and surely an oversimplification. Huge efforts have gone into modelling the complex nonlinear development of cortex, by teams including the developing Human Connectome Project. For such a fundamental aspect of this work, why isn't it possible to reconstruct the shape of this relatively small part of juvenile rat cortex?

• The same relative laminar depths are used for all subregions. This will have a large impact on the model. However, relative laminar depths can change drastically across the cortex (see e.g. many papers by Palomero-Gallagher, Zilles and colleagues). The authors should incorporate the real laminar depths, or, failing that, show evidence to show that the laminar depth differences across the subregions included in the model are negligible.

• The authors perform an affine mapping between mouse and rat cortex. This is again surprising. In human imaging, affine mappings are insufficient to map between two individual brains of the same species, and nonlinear transformations are instead used. That an affine transformation should be considered sufficient to map between two different species is then very surprising. For some models, this may be fine, but there is a supposed emphasis here on biological precision in terms of anatomical location.

o Live nature of the model. This is such a colossal model, and effort, that I worry that it may be quite difficult to update in light of new data. For example, how much person and compute time would it take to update the model to account for different layer sizes across subregions? Or to more precisely account for the shape of juvenile rat cortex?

---

## [Referee Report · Reviewer #3 (Public review)]

This manuscript reports a detailed model of the rat non-barrel somatosensory cortex, consisting of 4.2 million morphologically and biophysically detailed neuron models, arranged in space and connected according to highly sophisticated rules informed by diverse experimental data. Due to its breadth and sophistication the model will undoubtedly be of interest to the community, and the reporting of anatomical details of modeling in this paper is important for understanding all the assumptions and procedures involved in constructing the model. While a useful contribution to this field, the model and the manuscript could be improved by employing data more directly and comparing simple features of the model's connectivity - in particular, connection probabilities - with relevant experimental data.

The manuscript is overall well-written, but contains a substantial number of confusing or unclear statements, and some important information is not provided.

Comments on revisions:

The authors mostly addressed all my points and improved the paper substantially. I do not have further extensive comments except one general point below.

Regarding section 2.3 and metrics of connectivity like pairwise connection probabilities, it is great that the authors rewrote that section and added comparisons with experimental data in Figs. 4 and S9. Unfortunately, what one finds when direct comparisons are made is that the modeled pairwise connectivity is quite different from the data. Fig. S9 shows that the model's results do not agree with data in about half of the cases (purple and red arrows). Similarly large discrepancies can be seen for some other metrics, like in Fig. S10B and S10C1,C2. (And similar concerns apply to thalamocortical connections in section 2.5, where it looks like little to no data are available to verify the pairwise connectivity between the thalamic and cortical neurons via a direct comparison.)

This is concerning since this model forms the basis for multiple other studies of cortical dynamics and function by the same group and potentially others in the community, with multiple papers relying on it, whereas basic properties of connectivity are apparently not captured well.

On the other hand, this is also a "glass half full" situation, showing that the sophisticated algorithms for establishing connections, developed by the authors, are working well in at least half of the connection types explored. It is therefore imperative that the authors continue refining these algorithms to capture the remaining half in future iterations and producing improved models that the community can better rely on.

Please also note that Fig. S11 does not have a caption.

---

## [Author Response]

The following is the authors’ response to the original reviews.

**Public Reviews:**

**Reviewer #1 (Public Review):**
Summary:In this study, the authors describe the construction of an extremely large-scale anatomical model of juvenile rat somatosensory cortex (excluding the barrel region), which extends earlier iterations of these models by expanding across multiple interconnected cortical areas. The models are constructed in such a way as to maintain biological detail from a granular scale - for example, individual cell morphologies are maintained, and synaptic connectivity is founded on anatomical contacts. The authors use this model to investigate a variety of properties, from cell-type specific targeting (where the model results are compared to findings from recent large-scale electron microscopy studies) to network metrics. The model is also intended to serve as a platform and resource for the community by being a foundation for simulations of neuronal circuit activity and for additional anatomical studies that rely on the detailed knowledge of cellular identity and connectivity.Strengths:As the authors point out, the combination of scale and granularity of their model is what makes this study valuable and unique. The comparisons with recent electron microscopy findings are some of the most compelling results presented in the study, showing that certain connectivity patterns can arise directly from the anatomical configuration, while other discrepancies highlight where more selective targeting rules (perhaps based on molecular cues) are likely employed. They also describe intriguing effects of cortical thickness and curvature on circuit connectivity and characterize the magnitude of those effects on different cortical layers.The detailed construction of the model is drawn on a wide range of data sources (cellular and synaptic density measures, neuronal morphologies, cellular composition measures, brain geometry, etc.) that are integrated together; other data sources are used for comparison and validation. This consolidation and comparison also represent a valuable contribution to the overall understanding of the modeled system.

We thank the reviewer for the kind comments.

Weaknesses:The scale of the model, which is a primary strength, also can carry some drawbacks. In order to integrate all the diverse data sources together, many specific decisions must be made about, for example, translating findings from different species or regions to the modeled system, or deciding which aspects of the system can be assumed to be the same and which should vary. All these decisions will have effects on the predicted results from the model, which could limit the types of conclusions that can be made (both by the others and by others in the community who may wish to use the model for their own work).

We agree that this is a downside of the principle of biophysically detailed modeling that is best addressed by continuous refinement in collaboration with the community. We would like to once again invite any interested party to participate in this process.

As an example, while it is interesting that broad brain geometry has effects on network structure (Figure 7), it is not clear how those effects are actually manifested. I am not sure if some of the effects could be due to the way the model is constructed - perhaps there may be limited sets of morphologies that fit into columns of particular thicknesses, and those morphologies may have certain idiosyncrasies that could produce different statistics of connectivities where they are heavily used. That may be true to biology, but it may also be somewhat artifactual if, for example, the only neurons in the library that fit into that particular part of the cortex differ from the typical neurons that are actually found in that region (but may not have been part of the morphological sampling).

We agree that the limited pool of morphological reconstructions can lead to artifactual results in the way the reviewer pointed out. To investigate that hypothesis, we added a supplementary figure (S14) where we characterize (1): to what degree the morphological composition of a columnar subvolume reflects the overall composition of the model; and (2): The level of morphological diversity in each columnar subvolume. We discuss the results at the end of section 2.6. Briefly, while we cannot fully rule out the possibility of an artificial result, we found a high and virtually uniform level of morphological diversity in all columns and layers. This makes it unlikely that individual idiosyncratic morphologies strongly affect the local connectivity. However, we acknowledge that the minimum level of morphological diversity required is unknown. We believe that at this stage all we can do is characterize this and leave final interpretation to the reader.

I also wonder how much the assumption that the layers have the same relative thicknesses everywhere in the cortex affects these findings, since layer thicknesses do in fact vary across the cortex.

We agree that layer thickness variation would affect circuit properties. Variability of layer thickness can be split into two components: variability stemming from differences in total thickness, which our model covers, and variability of relative, i.e., normalized layer thickness, which we miss. In this region of cortex, though, data on the relative thickness of cortical layers is sparse. The Waxholm Atlas does not distinguish somatosensory cortical layers in its labels [Kleven et al, 2023]. Yusufoğulları (2015) compares layer thicknesses of rat hindlimb and barrel field regions. After normalization against total thickness, the relative difference increased towards the superficial layers from 0 in L6 to 33% in L1. Variability of normalized thicknesses within developed rat barrel cortex, based on layer boundaries reported in Narayanan et al. (2017) vary by 2% to 5% over approximately 2 mm. One major effect of such variability would be to scale the number of neurons in a given layer locally by the corresponding factors. For comparison, the resulting variability in neuron counts due to differences in conicality (Fig. 7D1) was around +-25%. A further effect of variable relative layer thickness would be its impact on the selection of suitable morphologies to be placed in the volume.

In summary, adjustment of layer thickness is a refinement which should be done in future versions of the model, once more data is available. The discussion section has been updated to acknowledge this limitation. However, as outlined at the beginning of this point-by-point reply, we will not conduct such updates to the model in the context of this manuscript, as it describes the version of the model used for a number of follow-up studies.

In addition, the complexity of the model means that some complicated analyses and decisions are only presented in this manuscript with perhaps a single panel and not much textual explanation. I find, for example, that the panels of Figure S2 seem to abstract or simplify many details to the point where I am not clear about what they are actually illustrating - how does Figure S2D represent the results of "the process illustrated in B"? Why are there abrupt changes in connectivity at region borders (shown as discontinuous colors), when dendrites and axons span those borders and so would imply interconnectivity across the borders? What do the histograms in E1 and E2 portray, and how are they related to each other?

We apologize for the confusion. We have updated the figure caption of Figure S2 to better explain its contents.

Overall, the model presented in this study represents an enormous amount of work and stands as a unique resource for the community, but also is made somewhat unwieldy for the community to employ due to the weight of its manifold specific construction decisions, size, and complexity.
**Reviewer #2 (Public Review):**
Summary:The authors build a colossal anatomical model of juvenile rat non-barrel primary somatosensory cortex, including inputs from the thalamus. This enhances past models by incorporating information on the shape of the cortex and estimated densities of various types of excitatory and inhibitory neurons across layers. This is intended to enable an analysis of the micro- and mesoscopic organisation of cortical connectivity and to be a base anatomical model for large-scale simulations of physiology.Strengths:• The authors incorporate many diverse data sources on morphology and connectivity.• This paper takes on the challenging task of linking micro- and mesoscale connectivity.• By building in the shape of the cortex, the authors were able to link cortical geometry to connectivity. In particular, they make an unexpected prediction that cortical conicality affects the modularity of local connectivity, which should be testable.• The author's analysis of the model led to the interesting prediction that layer 5 neurons connect local modules, which may be testable in the future, and provide a basis to link from detailed anatomy to functional computations.• The visualisation of the anatomy in various forms is excellent.• A subnetwork of the model is openly shared (but see question below).

We thank the reviewer for their kind comments.

Weaknesses:• Why was non-barrel S1 of the juvenile rat cortex selected as the target for this huge modelling effort? This is not explained.

We have added an explanation of this decision to the third paragraph of the introduction.

• There is no effort to determine how specific or generalisable the findings here are to other parts of the cortex. Although there is a link to physiological modelling in another paper, there is no clear pathway to go from this type of model to understand how the specific function of the modelled areas may emerge here (and not in other cortical areas).

With respect to generality against specific findings, our philosophy is as follows: Despite the fact that most of our source data comes from juvenile rat somatosensory cortex, we also had to generalize many data sources across organisms, ages or regions. Hence, in this iteration we focused on investigating the general features of the (multi-region) mammalian cortex, e.g., high-order motifs, connected by L5 neurons across subregions or the effect of curvature on the connectivity. In the future, more specific data sources can be used to build diverging versions of the model, e.g. one for adult vs. juvenile rat. They can then be used to contrast the ages and focus on more specific findings. We already defined a number of structural metrics that can be used to contrast more specific versions of the model quantitatively.

We now clarify this pathway to understanding more specific function in the last paragraph of the discussion.

• In a few places the manuscript could be improved by being more specific in the language, for example:- "our anatomy-based approach has been shown to be powerful", I would prefer instead to read about specific contributions of past papers to the field, and how this builds on them.- similarly: "ensuring that the total number of synapses in a region-to-region pathway matches biology." Biology here is a loose term and implies too much confidence in the matching to some ground truth. Please instead describe the source of the data, including the type of experiment.

We have removed or rewritten the mentioned parts. We now clarify that we work based on biological estimates from experiments and cite the experiment sources. We also provide brief descriptions of the types of data and how they were derived.

• Some of the decisions seem a little ad-hoc, and the means to assess those decisions are not always available to the reader e.g.- pg. 10. "Based on these results, we decided that the local connectome sufficed to model connectivity within a region.". What is the basis for this decision? Can it be formalised?- "In the remaining layers the results of the objective classification were used to validate the class assignments of individual pyramidal cells. We found the objective classification to match the expert classification closely (i.e., for 80-90% of the morphologies). Consequently, we considered the expert classification to be sufficiently accurate to build the model." The description of the validation is a little informal. How many experts were there? What are their initials? Was inter-rater or intra-rater reliability assessed? What are these numbers? The match with Kanari's classification accuracy should be reported exactly. There are clearly experts among the author list, but we are all fallible without good controls in place, and they should be more explicit about those controls here, in my opinion.- "Morphology selection was then performed as previously (Markram et al., 2015), that is, a morphology was selected randomly from the top 10% scorers for a given position." A lot of the decisions seem a little ad-hoc, without justification other than this group had previously done the same thing. For example, why 10% here? Shouldn't this be based on selecting from all of the reasonable morphologies?

We have clarified that the density of local connectivity is verified against the validation datasets by comparing the diagonals in Figure 4B, in addition to the quantification of Figure 4C.

For the classification, we have now published a detailed preprint describing the objective confirmation of expert classification by a variety of methods (see Kanari et al. 2024 https://www.biorxiv.org/content/10.1101/2024.09.13.612635v1). We cannot include the full methodology in the current paper, due to its large extent. For the benefit of the reader, we have included the appropriate citation and extended the short description of the methodology. As described in this paper, the classification accuracy varies per layer, cell type, etc. We have now described in more details these results, that can be accessed in details in out preprint.

• I would like to know if one of the key results relating to modularity and cortical geometry can be further explored. In particular, there seem to be sharp changes in the data at the end of the modelled cortical regions, which need to be explored or explained further.

We now explore these results further in supplementary figure S15, which we discuss in the results Section 2.6.

• The shape of the juvenile cortex - a key novelty of this work - was based on merely a scalar reduction of the adult cortex. This is very surprising, and surely an oversimplification. Huge efforts have gone into modelling the complex nonlinear development of the cortex, by teams including the developing Human Connectome Project. For such a fundamental aspect of this work, why isn't it possible to reconstruct the shape of this relatively small part of the juvenile rat cortex?

We agree that a more complex approach should be used in the future. However, as outlined at the beginning of this point-by-point reply, we will not conduct such updates to the model in the context of this manuscript, as it describes the version of the model used for a number of follow-up studies.

• The same relative laminar depths are used for all subregions. This will have a large impact on the model. However, relative laminar depths can change drastically across the cortex (see e.g. many papers by Palomero-Gallagher, Zilles, and colleagues). The authors should incorporate the real laminar depths, or, failing that, show evidence to show that the laminar depth differences across the subregions included in the model are negligible.

This point has also been raised by reviewer #1 above. For convenience, we repeat our reply below.

We agree that layer thickness variation would affect circuit properties. Variability of layer thickness can be split into two components: variability stemming from differences in total thickness, which our model covers, and variability of relative, i.e., normalized layer thickness, which we miss. In this region of cortex, though, data on the relative thickness of cortical layers is sparse. The Waxholm Atlas does not distinguish somatosensory cortical layers in its labels [Kleven et al, 2023]. Yusufoğulları (2015) compares layer thicknesses of rat hindlimb and barrel field regions. After normalization against total thickness, the relative difference increased towards the superficial layers from 0 in L6 to 33% in L1. Variability of normalized thicknesses within developed rat barrel cortex, based on layer boundaries reported in Narayanan et al. (2017) vary by 2% to 5% over approximately 2 mm. One major effect of such variability would be to scale the number of neurons in a given layer locally by the corresponding factors. For comparison, the resulting variability in neuron counts due to differences in conicality (Fig. 7D1) was around +-25%. A further effect of variable relative layer thickness would be its impact on the selection of suitable morphologies to be placed in the volume.

In summary, adjustment of layer thickness is a refinement which should be done in future versions of the model, once more data is available. The discussion section has been updated to acknowledge this limitation. However, as outlined at the beginning of this point-by-point reply, we will not conduct such updates to the model in the context of this manuscript, as it describes the version of the model used for a number of follow-up studies.

• The authors perform an affine mapping between mouse and rat cortex. This is again surprising. In human imaging, affine mappings are insufficient to map between two individual brains of the same species and nonlinear transformations are instead used. That an affine transformation should be considered sufficient to map between two different species is then very surprising. For some models, this may be fine, but there is a supposed emphasis here on biological precision in terms of anatomical location.

We agree that this is a weakness that we will address in future revisions of the model.

• One of the most interesting conclusions, that the connectivity pattern observed is in part due to cooperative synapse formation, is based on analyses that are unfortunately not shown.

We originally decided not to show this part as we underestimated the interest in this particular result. We have now included the result in supplementary figure S10 and discuss the figure in the results.

• Open code:- Why is only a subvolume available to the community?

We have now made the entire model available under doi.org/10.7910/DVN/HISHXN. The Data and Code availability section has been updated to clarify this.

- Live nature of the model. This is such a colossal model, and effort, that I worry that it may be quite difficult to update in light of new data. For example, how much person and computer time would it take to update the model to account for different layer sizes across subregions? Or to more precisely account for the shape of the juvenile rat cortex?

To provide more information to people interested in participating in model refinements, we have added a new Figure 9. We discuss potential opportunities for refinement at the end of the discussion section.

**Reviewer #3 (Public Review):**
This manuscript reports a detailed model of the rat non-barrel somatosensory cortex, consisting of 4.2 million morphologically and biophysically detailed neuron models, arranged in space and connected according to highly sophisticated rules informed by diverse experimental data. Due to its breadth and sophistication, the model will undoubtedly be of interest to the community, and the reporting of anatomical details of modeling in this paper is important for understanding all the assumptions and procedures involved in constructing the model. While a useful contribution to this field, the model and the manuscript could be improved by employing data more directly and comparing simple features of the model's connectivity - in particular, connection probabilities - with relevant experimental data.The manuscript is well-written overall but contains a substantial number of confusing or unclear statements, and some important information is not provided.Below, major concerns are listed, followed by more specific but still important issues.Major issues(1) Cortical connectivity.Section 2.3, "Local, mid-range and extrinsic connectivity modeled separately", and Figure 4: I am confused about what is done here and why. The authors have target data for connectivity (Figure 4B1). But then they use an apposition-based algorithm that results in connectivity that is quite different from the data (Figure 4B2, C). They then use a correction based on the data (Figure 4E) to arrive at a more realistic connectivity. Why not set the connectivity based on the data right away then? That would seem like a more straightforward approach.

We have completely re-written our description and discussion of connectivity in the model. We now more explicitly motivate our connectivity modeling choices in the first paragraph of section 2.3 of the results and in the second paragraph of the discussion.

The same comment applies to Section 2.4., "Specificity of axonal targeting": the distributions of synapses on different types of target cell compartments were not well captured by the original model based on axon-dendrite overlap and pruning, so the authors introduced further pruning to match data specificity. While details of this process and what worked and what didn't may be interesting to some, overall it is not surprising, as it has been well known that cell types exhibit connectivity that is much more specific than "Peters rule" or its simple variations. The question is, since one has the data, why not use the data in the first place to set up the connectivity, instead of using the convoluted process of employing axon-dendrite overlap followed by multiple corrections?

We would like to point out that we are not employing “Peters rule”, we now make this explicit in the revision in the first paragraph of section 2.3 of the results. Furthermore, we would argue that the match to the Motta et al. data indicates that our approach is more than just a “simple variation”. Finally, we believe that there is important insight in: 1. The specific ways in which the algorithm had to be changed to match the Schneider-Mizell data, e.g. that the connectivity of SST positive neurons did not have to be adapted at all. 2. That the specificity of the other two types could still be matched by a selection of a subset of axonal appositions (i.e., of potential synapses).

Most importantly, what is missing from the whole paper is the characterization of connection probabilities, at least for the local circuit within one area. Such connection probabilities can be obtained from the data that the authors already use here, such as the MICRONS dataset. Another good source of such data is Campagnola et al., Science, 2022. Both datasets are for mouse V1, but they provide a comprehensive characterization across all cortical layers, thus offering a good benchmark for comparison of the model with the data. It would be important for the authors to show how connection probabilities realized in their model for different cell types compared to these data.

We now report connection probabilities in the reworked figure 4 and compare them to reported connection probabilities from many different sources and labs in supplementary figure S8. We prefer a comparison to a wide range of sources to relying on a single report.

(2) Section 2.5, "Structure of thalamic inputs" and Figure 6.The text in section 2.5 should provide more details on what was done - namely, that the thalamic axons were generated based on the axon density profiles and then synapses were established based on their overall with cortical dendrites. Figure S10 where the target axon densities from data and the model axon densities are compared is not even mentioned here. Now, Figure S10 only shows that the axon densities were generated in a way that matches the data reasonably well. However, how can we know that it results in connectivity that agrees with data? Are there data sources that can be used for that purpose? For example, the authors show that in their model "the peaks of the mean number of thalamic inputs per neuron occur at lower depths than the peaks of the synaptic density". Is this prediction of the model consistent with any available data?Most importantly, the authors should show how the different cell types in their model are targeted by the thalamic inputs in each layer. Experimental studies have been done suggesting specificity in targeting of interneuron types by thalamic axons, such as PV cells being targeted strongly whereas SST and VIP cells being targeted less.

We have updated the Results section to provide context for the thalamic axon placement, and referred the reader to the methods for more detail. A reference to Figure S10 has now been added to this section as well.

As for validations of the structure of the thalamo-cortical inputs: We found that the existing literature on the topic, such as Cruikshank et al., 2007, 2010 and more recently Sermet et al., 2019, is predominately on the physiological strengths of the pathways. We acknowledge that the authors provide compelling arguments that their findings are likely partially due to differences in the anatomical innervation strengths. On the other hand, Sporns, 2013 cautioned against mixing up structural and functional connectivity. Overall, we believe that it is simply cleaner to perform this validation in the accompanying manuscript (“Part II: Physiology and Experimentation”), using the full physiological model. Note that we have actually performed that validation in the manuscript (see preprint under the following doi: 10.1101/2023.05.17.541168, Figure 3H1).

Note that a higher physiological strength onto PV+ neurons is observed.

(3) "We have therefore made not only the model but also most of our tool chain openly available to the public (Figure 1; step 7)."In fact it is not the whole model that is made publicly available, but only about 5% of it (211,000 out of 4,200,000 neurons). Also, why is "most" of the tool chain made openly available, and not the whole tool chain?

We have now made the entire model available under doi.org/10.7910/DVN/HISHXN. This has also been added to the Key resource table.

With regard to the tool chain, everything is on our public github (https://github.com/BlueBrain/) except for the algorithm for detecting axonal appositions. For that tool there are currently unresolved potential copyright issues with former collaboration partners. We are working to resolve them.

Other issues"At each soma location, a reconstruction of the corresponding m-type was chosen based on the size and shape of its dendritic and axonal trees (Figure S6). Additionally, it was rotated to according to the orientation towards the cortical surface at that point."After this procedure, were cells additionally rotated around the white matter-pia axis? If yes, then how much and randomly or not? If not, then why not? Such rotations would seem important because otherwise additional order potentially not present in the real cortex is introduced in the model affecting connectivity and possibly also in vivo physiology (such as the dynamics of the extracellular electric field).

They are indeed additionally randomly rotated. We have clarified this in the revision.

The term "new in vivo reconstructions" for the 58 neurons used in this paper in addition to "in vitro reconstructions" is a misnomer. It is not straightforward to see where the procedure is described, but then one finds that the part of Methods that describes experimental manipulations is mostly about that (so, a clearer pointer to that part of Methods could be useful). However, the description in Methods makes it clear that it is only labeling that is done in vivo; the microscopy and reconstruction are done subsequently in vitro. I would recommend changing the terminology here, as it is confusing. Also, can the authors show reconstructions of these neurons in the supplementary figures? Is the reconstruction shown in Figure 4A representative?

The term is used because the staining is done in vivo. To the best of our knowledge, the reconstruction process cannot be performed in vivo. However, to avoid any confusion we modified the text to clarify this distinction to in-vivo stained.

With respect to the reconstruction in Figure 4: The intent of the panel is to demonstrate the concept of targeted long-range axons that our morphologies are missing, necessitating the use of a second algorithm for longer-range connectivity. As such, it is not one of the reconstructions we used, but one of Janelia MouseLight. While we mentioned MouseLight in the figure caption, we formulated it in a way that could be misunderstood to mean that we merely used the MouseLight browser to render one of our morphologies. We apologize for the confusion, and we have fixed the figure caption.

In this revision we have added exemplars of representative morphology reconstructions (in slice stained and in vivo stained) in a new supplementary figure, as requested (Figure S5). It is referenced in the last paragraph of section 2.1.

In the Discussion, "This was taken into account during the modeling of the anatomical composition, e.g. by using three-dimensional, layer-specific neuron density profiles that match biological measurements, and by ensuring the biologically correct orientation of model neurons with respect to the orientation towards the cortical surface. As local connectivity was derived from axo-dendritic appositions in the anatomical model, it was strongly affected by these aspects.However, this approach alone was insufficient at the large spatial scale of the model, as it was limited to connections at distances below 1000μm."As mentioned above, it is not clear that this approach was sufficient for local connectivity either. It would be great if the authors showed a systematic comparison of local connection probabilities between different cell types in their model with experimental data and commented here in the Discussion about how well the model agrees with the data.

As mentioned in the reply to a previous comment, we now report connection probabilities.

In the Discussion: "The combined connectome therefore captures important correlations at that level, such as slender-tufted layer 5 PCs sending strong non-local cortico-cortical connections, but thick-tufted layer 5 PCs not." (Also the corresponding findings in Results.)If I understand this statement correctly, it may not agree with biological data. See analysis from MICRONS dataset in Bodor et al., https://www.biorxiv.org/content/10.1101/2023.10.18.562531v1.

Our statement was indeed misleading and formulated too strongly. While thick-tufted pyramidal cells do form long-range intra-cortical connections, the structural strength of these pathways is weaker than for slender-tufted PCs, which are associated with the IT (intra-telencephalic) projection type. We have made this clear in the revision.

Table 2 is confusing. What do pluses and minuses mean? What does it mean that some entries have two pluses? This table is not mentioned anywhere else in the text. If pluses mean some meaningful predictions of the model, then their distribution in the table seems quite liberal and arbitrary. It is not clear to me that the model makes that many predictions, especially for type-specificity and plasticity. Also, why is the hippocampus mentioned in this table? I don't see anything about the hippocampus anywhere else in the paper.

We have clarified the description of the table in its caption and removed references to hippocampus, which were left from an earlier draft of the paper.

In the Discussion, "Thus, we made the tools to improve our model also openly available (see Data and Code availability section)."As mentioned before, the authors themselves write that they made "most of our tool chain openly available to the public", but not all of it.

With regard to the tool chain, everything is on our public github (https://github.com/BlueBrain/) except for the algorithm for detecting axonal appositions. For that tool there are currently unresolved potential copyright issues with former collaboration partners. We are working to resolve them.

Table S2 has multiple question marks. It is not clear whether the "predictions" listed in that table are truly well-thought-out and/or whether experimental confirmations are real.

Some of the citations in that table were broken due to technical difficulties with the citation manager used. We apologize and have fixed this in the revision.

Introduction: It would be quite appropriate to cite here Einevoll et al., Neuron, 2019 ("The Scientific Case for Brain Simulations").

We now reference this important work.

**Recommendations for the authors:**

**Reviewing Editor's note:**
Consultation with the reviewers highlighted three main issues: the integration of connection probability profiles, non-uniform cortical thickness, and the overall organization of the manuscript.
**Reviewer #1 (Recommendations For The Authors):**
Apart from the points discussed in the public review, my main concern is that the manuscript itself is not as tightly constructed as it should be, to the detriment of the reader's ability to understand the model itself and the conclusions from the presented analyses.There are places where the text references seemingly incorrect figure panels or refers to panels that don't exist:- Section 2.2, first paragraph - refers to Figure 2D, E but those panels do not exist in Figure 2.- Section 2.2, second paragraph - refers to Figure 3D3 - perhaps it should be 3B3?- Section 2.8, first paragraph - has no figure references but seems like it should be referring to parts of Figure 8 (perhaps Figure 8B1 specifically?)- Is the reference to Figure S11A on page 16 supposed to be to S12A?In other places, figure labels and descriptions are not clear, and terminology is not always well-defined or explained.- Figure 8 and the associated section 2.8 are very difficult to draw conclusions from as presented - several of the terms used are opaque and not clearly defined in the text or legends. I could not easily infer how the normalization works for the "normalized node participation per layer", or what "position in simplex" means for "unique neurons in core", and what their "relative counts" are relative to.- Are "targets" in Figure S12A the same as "sinks"? If so, it would be better to use a single term consistently throughout.- Figure S12 - figures in part B do not have enough labels to interpret - what is the y-axis of the "rich-club analysis" graph? Also, the figures in part B bottom are labeled "long-range" rather than "mid-range" connections.In general, I found the use of both letters and numbers for figure panels (e.g. Figure 7E1) more confusing than helpful - it didn't seem like panels with the same letter were visually grouped consistently, and it sometimes made it more difficult to follow the flow of a figure. I would recommend using only letters in nearly every case here.

We thank the reviewer for directing our attention to these issues. We have fixed them in the revision. However, we have decided to keep our original panel numbering scheme. Panels with the same letter are meant to be conceptually grouped as they address related or similar measures.

Other minor points:- Section 2.4 - paragraph 2 - sentence 5 "inhbititory" -> "inhibitory".- Figure 5B figure legend - references Schneider-Mizell et al. 2023 but probably should be Motta et al. 2019?- Figure 5C - figure key "expcected" -> "expected".- The lower part of Figure 7C looks like it belongs to panel D2 instead of panel C due to relative spacing.

We once again thank the reviewer, and we have fixed the listed issues.

**Reviewer #2 (Recommendations For The Authors):**
(1) Abstract:- Is it really 'integrating whole brain-scale data'? This seems a bit misleading.- "We delineated the limits of determining connectivity from anatomy" - here I think you mean determining connectivity from morphology, or dendrite/axon appositions. Electron microscopy is still anatomy and presumably would be much closer to function.

We originally used the term “anatomy” as connectivity depends on the correct placement of neurons in addition to their morphology. However, as the reviewer points out, this term is misleading as it would encompass electron microscopy, which can go beyond what we do with the model. We have updated the text to read “morphology and placement”.

(2) Introduction:"Investigating the multi-scale interactions that shape perception requires a model of multiple cortical subregions with inter-region connectivity, but it also requires the subcellular resolution provided by a morphologically detailed model." - This statement, as written, is not true in my opinion. You can argue for the value of morphologically-detailed neuron models to the study of perception, but they are not required for the investigation of perception.

We have updated the text to be clearer: subcellular resolution is only required for certain aspects that are related to perception.

(3) Results:- Pg. 9/10. There are three sentences in a row that are of the style: "ensuring that the total number of synapses in a region-to-region pathway matches biology." Biology here is a loose term and implies too much confidence in the matching to some ground truth. Please instead describe the source of the data, including the type of experiment here already. o Pg. 10. On the first read, I found it quite hard to follow what exactly was done in Figure 4.What are the target values adapted from Reimann et al., 2019, for example?- Pg. 10. "Based on these results, we decided that the local connectome sufficed to model connectivity within a region.". What is the basis for this decision? Can it be formalised? o Pg. 16, Figure 7 B-C. The apparent effect of geometry on modularity is potentially very interesting. However, are the sharp drop-offs in values for modularity (but also conicality and height) true, or are some artefacts due to columns at the edges of the sampled area?

We have discussed these points above in the general comments and strengths and weaknesses.

- Pg. 18. Simplicial cores define central subnetworks, tied together by mid-range connections. This work, in particular leading to the conclusion of the layer 5 highway hubs, stands out as being a successful attempt to simplify the highly detailed model to a degree that it generates useable new understanding.

We thank the reviewer for the kind comment.

(4) Figures:Figure 2: The caption doesn't seem to match the Figure (e.g. there are no brain regions depicted in A). o Figure 4f. This is a key panel, but is squished into a small corner of Figure 4, and therefore hard-to-read.

We have fixed this in the revision.

**Reviewer #3 (Recommendations For The Authors):**
In Major comments, point (1) discusses the issue of connectivity known from data. For all the aspects of connectivity mentioned there, I would recommend the authors re-build their model using the connectivity data directly. It would be interesting to test whether a model constructed in such a way would have any difference in simulated neural activity relative to the model they have constructed.

This is indeed a very interesting avenue of research. However, we believe that it is best conducted in separate manuscripts. First, in Pokorny et al., 2024 (https://doi.org/10.1101/2024.05.24.593860) we conduct this investigation, comparing the emerging activity in the model to the one for simpler connectivity models. Additionally, in Egas-Santander et al., 2024 (https://www.biorxiv.org/content/10.1101/2024.03.15.585196v3) we found that simpler connectomes lead to less reliable spiking activity globally. Finally, in the accompanying manuscript (https://www.biorxiv.org/content/10.1101/2023.05.17.541168v5) we compare activity with and without the targeting specificity of Schneider-Mizell et al.

In Major comments, point (2) discusses thalamic inputs. I would recommend the authors to address the issues mentioned there.

We have replied to those comments above.

In addition, panels F and G of Figure 6 are mentioned in the caption but are not shown in the figure. In panel B, the choice of visualization is strange. It would make sense to show box plots for all the data instead of bars for mean values and points for randomly selected 50 cells. Panels E1 and E2 lack units.

We have removed mentions of panels F and G and changed the style of plot. Units for E1 and E2 are now explained in the figure caption.

In Major comments, point (3) touches upon model and tool sharing. I would recommend making such statements more accurate and reflecting what exactly is provided to the community since not everything is shared.

We have now made the entire model available under doi.org/10.7910/DVN/HISHXN.

With regard to the tool chain, everything is on our public github (https://github.com/BlueBrain/) except for the algorithm for detecting axonal appositions. For that tool there are currently unresolved potential copyright issues with former collaboration partners. We are working to resolve them.

I would recommend the authors address all the other points mentioned in the public review as well. In addition, below are some smaller issues that should be fixed.Figure 2: the caption appears to be partially wrong and partially misassigned to the figure panels.

We fixed the issue.

Also, note that in L6 the types L6_TPC:A and L6_TPC:C are listed in the figure, but L6_TPC:B is not mentioned.

There is indeed no TPC:B type in layer 6. The distinction between TPC:A and TPC:B is based on early or late bifurcations of the apical dendrite and is only observed in layer 5.

Figure 3, panel B2: the caption refers to colors in panel (C), but the authors probably meant to refer to panel (A).

We fixed the issue.

"The placement of morphological reconstructions matched expectation, showing an appropriately layered structure with only small parts of neurites leaving the modeled volume (Figure 2D, E)."Figure 2 does not have panels D and E."The volume was clearly dominated by dendrites, filling between 23% and 47% of the space, compared to 2% to 11% for axons (Figure 3D3)." There is no panel D or D3 in Figure 3."Recently, the MICrONS dataset (MICrONS-Consortium et al., 2021) has been analyzed with respect to the axonal targeting of inhibitory subtypes in a 100 x 100 μm subvolume spanning all layers (Schneider-Mizell et al., 2023)."100 x 100 μm is an area (and should be 100 x 100 μm^2), not a volume.Figure S11B requires a legend for the color map.

We fixed the issues.

Table S1: What is the difference between L6_BP and L6_BPC? They both are referred to as L6 bipolar cells.

We have changed the description of L6_BPC to “Layer 6 bitufted pyramidal cell”.